# Enhanced virulence and stress tolerance are signatures of epidemiologically successful *Shigella sonnei*

Sydney L. Miles[1], Dilys Santillo[2], Hannah Painter[1], Kathryn Wright[1], Vincenzo Torraca[1,12], Ana T. López-Jiménez[1], Mollie Virgo[1], Xosé M. Matanza[3], Abigail Clements[3], Claire Jenkins[4], Stephen Baker[5], Kate S. Baker[6], David Cisneros[7], Andrea Puhar[7,8], Vanessa Sancho-Shimizu[2,9,10], Kathryn E. Holt[1,11] & Serge Mostowy[1]✉

Shigellosis is a leading cause of diarrhoeal deaths, with *Shigella sonnei* increasingly implicated as a dominant agent. *S. sonnei* is divided into five monophyletic lineages, yet most infections are caused by a few clonal sub-lineages within Lineage 3 that are quite distinct from the widely used Lineage 2 laboratory strain 53G. Factors underlying the success of these globally dominant lineages remain unclear in part due to a lack of complete genome sequences and animal models. Here, we utilise a novel reference collection of representative Lineage 1, 2 and 3 isolates and find that epidemiologically successful *S. sonnei* harbour fewer genes encoding putative immunogenic components whilst key virulence-associated regions (including the type three secretion system and O-antigen) remain highly conserved. Using a zebrafish infection model, Lineage 3 isolates proved most virulent, driven by increased dissemination and a greater neutrophil response. These isolates also show increased resistance to complement-mediated killing alongside upregulated expression of group four capsule synthesis genes. Consistently, primary human neutrophil infections revealed an increased tolerance to phagosomal killing. Together, our findings link the epidemiological success of *S. sonnei* to heightened virulence and stress tolerance, and highlight zebrafish as a valuable platform to illuminate factors underlying establishment of epidemiological success.

Shigella represents a group of human-adapted lineages of *Escherichia coli* that have independently evolved to cause severe diarrhoeal disease. Each subgroup emerged following the acquisition of a large virulence plasmid (pINV), which encodes a type three secretion system (T3SS)[1]. There are four recognised subgroups of *Shigella*: *S. boydii, S. dysenteriae, S. flexneri* and *S. sonnei*, but most contemporary infections are caused by *S. flexneri* and *S. sonnei*[2]. For many years, *S. flexneri* was regarded as the leading cause of shigellosis in low- and middle-income countries, whilst *S. sonnei* dominated in high-income countries[3]. However, in the last two decades, an intercontinental shift has been recorded, with relative proportions of *S. sonnei* replacing *S. flexneri* in economically transitioning countries[4]. The drivers of this shift remain unclear, but proposed explanations include differences in O-antigen structure, the ability to acquire antimicrobial resistance (AMR) and bacterial competition strategies[4–6].

Within *S. sonnei*, there are five distinct lineages (1–5, each defined by ~600 single-nucleotide variations), which have undergone varying degrees of global dissemination[7,8]. Lineage 3 represents the most frequently detected lineage, with Clades 3.6 (Central Asia/CipR) and 3.7 (Global 3) dominating the epidemiological landscape[9]. Genome

reduction is a common feature of many major human-restricted pathogens, including *Mycobacterium tuberculosis, Bordetella pertussis* and *Salmonella enterica* serovar Typhi[10–12]. This phenomenon has been documented in all *Shigella* subgroups[13], but due to the instability of *S. sonnei* pINV in culture[14] it has not been fully explored in this species. We recently generated closed genome sequences of 15 diverse *S. sonnei* lineage 1-3 isolates and found that Lineage 3 chromosomes and pINV plasmids were most reduced[15]. The acquisition of stable, chromosomally encoded AMR determinants[7,16] and the presence of interbacterial competition weapons, such as colicins[17,18], have been linked to the epidemiological success of subgroups of Lineage 3 *S. sonnei*. However, most previous findings have been inferred from in silico analysis, and functional variations in infection models have not been widely explored, limiting our understanding of key epidemiological drivers.

*S. sonnei* is notorious for high levels of pINV instability during laboratory culture, which renders it avirulent[14]. Consequently, *S. flexneri* has been the preferred model for *Shigella* infection for many decades[19,20]. When *S. sonnei* has been used, infection models have mostly been restricted to a single lab-adapted strain first isolated in 1954 (53G, belonging to Lineage 2), which is phylogenetically distinct from currently circulating isolates[21]. As a result, our understanding of *S. sonnei* virulence has been mostly extrapolated from *S. flexneri* models. However, recent studies have highlighted key differences between the two subgroups[22], including reduced induction of macrophage pyroptosis by *S. sonnei*[23], increased tolerance against neutrophil-mediated killing[5] and the presence of a functional group four capsule (G4C) in *S. sonnei*, which has been shown to modulate its pathogenesis[24].

Considering this, along with a shift towards *S. sonnei* dominance and the upsurge in multidrug-resistant (MDR) infections, there is a clear need for further development and use of *S. sonnei* infection models. Mice are naturally resistant to *Shigella* infection, partly due to differences in inflammasome biology, and the recently established NAIP-NLRC4-deficient mouse model has been used to model *S. flexneri* infections exclusively so far[25]. The zebrafish infection model has therefore emerged as a powerful tool, recapitulating many key aspects of the *Shigella* infection process[26–28]. In the last decade, the zebrafish model has transformed our understanding of the cell biology[29,30] and host immune response[5,31,32] to *Shigella* infection, as well as revealed mechanisms underlying persistent infections in humans[33,34].

To investigate factors driving the rapid success of Lineage 3 *S. sonnei*, here we analyse our recently described reference collection of completely sequenced, epidemiologically relevant *S. sonnei* isolates[15]. Genomic comparisons reveal the specific loss of genes encoding putative immunogenic components. Zebrafish infection of the cognate isolates demonstrate that Lineage 3 strains are significantly more virulent in vivo, owing to increased in-host dissemination and a greater neutrophil response compared to Lineage 2. In agreement with in vitro results showing increased stress tolerance and expression of G4C, infection of primary human neutrophils shows that Lineage 3 isolates exhibit greater tolerance to phagosomal killing. Thus, our work delivers a deep understanding of currently circulating *S. sonnei*, highlighting enhanced virulence and stress tolerance as signatures of epidemiological success in this major human pathogen.

## Results

### Lineage 3 *S. sonnei* harbour fewer genes encoding immunogenic components

Previous studies have demonstrated an association between bacterial virulence and genome reduction[35,36]; to test for this in *S. sonnei*, we screened our novel reference collection for previously defined *E. coli* and *Shigella* virulence-associated genes (Table 1)[37]. Lineage 3 genomes harboured the fewest, at 80 per genome, whilst Lineage 1 and 2 genomes harboured between 89 and 91 (Fig. 1). In agreement with a

fundamental role for T3SS-mediated virulence in *S. sonnei*[23], the presence of T3SS-associated genes was highly conserved between all isolates. We found that most variation in virulence-associated genes was in the number associated with immunogenic components (such as adhesins and fimbriae), with Lineage 3 genomes harbouring only 3 such genes (remnants of type 1 (*fim*) and curli (*csg*) fimbrial loci), in contrast to the 12–14 genes carried in Lineage 1 and 2 genomes (including more complete *fim* and *csg* loci, as well as K88 (*fae*) and Afa/Dr (*afa*) fimbrial loci and the secreted autotransporter toxin *sat*) (Fig. 1). Considering that *S. flexneri* has been suggested to express adherence genes despite pseudogenisation of some genes within associated operons (including *fimA*, *csgA* and *csgB*)[38], we manually inspected the relevant genomic regions. We observed that some components of the *fim* locus (*fimH, fimF, fimD, fimG* and *fimB*) were present in Lineage 1 and 2 genomes. However, this ~9.3 kb region, located between *gntP* and *nanC*, was absent from all Lineage 3 genomes (although a *fimB* homologue was detected elsewhere in the genomes). In addition, *fimA* (encoding the major subunit) and *fimC* (encoding the chaperone) were not detected in any of the genomes analysed, consistent with previous reports for *S. sonnei*[39]. For curli, we noted that other curli subunits (*csgA, csgC, csgE, csgF* and *csgG*) were present and intact, making expression possible, albeit unlikely, since *csgD*, which acts as the master regulator[40], is interrupted by an IS600 insertion at approximately 0.18 kilobases from the start of the gene in both Lineages 2 and 3 genomes. Based on these genetic differences, we predict that Lineage 3 isolates lack *sat* as well as type 1 and *K88* fimbriae, which are recognised as highly immunogenic[41,42], and hypothesise that Lineage 3 may benefit from evading or tolerating the host immune response.

### Lineage 3 *S. sonnei* are more virulent in vivo

We selected four clinical isolates to test in the zebrafish infection model: Lineage 1.5, the well-characterised Lineage 2.8 (53G) and representatives from epidemiologically significant Clades 3.6 and 3.7 (Table 1). Zebrafish larvae were infected with 2000 colony forming units (CFU) in the hindbrain ventricle (HBV) and incubated at the standard temperature of 28.5 °C, where there were no significant differences in virulence (Fig. 2a). Given the previously demonstrated importance of thermoregulated virulence in *Shigella*[43] we next performed infections at 32.5 °C, a temperature which both activates the T3SS and has been shown to have no impact on zebrafish mortality[5]. Here, the survival of larvae infected with Lineage 3 isolates reduced to ~30%, significantly less than larvae infected with Lineage 1.5 or 2.8 (Fig. 2b). To confirm this finding, additional representatives from each lineage were tested, and in all cases, Clade 3.6 and 3.7 isolates were more virulent (Fig. S1a–d). To test whether Lineage 3 isolates had a replicative advantage in vivo, bacterial burden was enumerated from infected larvae at 6 and 24 h post-infection (hpi). This revealed a higher bacterial load when larvae were incubated at 32.5 °C, but differences were not lineage-dependent at either time-point (Fig. 2c, d). Considering there were no differences between Lineage 1.5 and Lineage 2.8, subsequent experiments were carried out using the well-characterised 53G as a comparison for Lineage 3.

Recent studies using zebrafish infection demonstrated that *S. sonnei* 53G disseminates from the HBV down the neural tube at a much higher frequency than *S. flexneri* serotype 5a strain M90T[44]. To investigate if dissemination from the HBV varies between lineages of *S. sonnei*, infections were performed with fluorescently labelled bacteria; larvae were imaged and then assessed for dissemination events. Dissemination was observed for all isolates tested, but increased dissemination was recorded for Clades 3.6 and 3.7 relative to Lineage 2.8 (chosen to represent non-Lineage 3 isolates) (Fig. 2e, f), consistent with an increased propensity to subvert the immune response.

**Table 1 | Details of bacterial strains used in this study**

| Species | Strain ID | Year of isolation | Country of Isolation | Description | Serotype/ genotype | Genome accession | Source |
|---|---|---|---|---|---|---|---|
| Shigella sonnei | 53G | 1954 | Japan | Lab strain | 2.8 | NC_016822 | 5 |
| Shigella sonnei | 53G Δwaal | NA | NA | O antigen mutant, Carb$^R$, Kan$^R$ | 2.8 | NA | 5 |
| Shigella sonnei | 53G Δg4c | NA | NA | Capsule mutant, Carb$^R$ | 2.8 | NA | 5 |
| Shigella sonnei | 53G GFP | NA | NA | Transformed with pFPV25.1, GFP reporter, Carb$^R$ | 2.8 | NA | 5 |
| Shigella sonnei | 02-1157 GFP | NA | NA | Transformed with pFPV25.1, GFP reporter, Carb$^R$ | 3.6.1.1.1 | NA | Generated in this study |
| Shigella sonnei | 03-0142 GFP | NA | NA | Transformed with pFPV25.1, GFP reporter, Carb$^R$ | 3.7.29.1.4 | NA | Generated in this study |
| Shigella sonnei | 201809330 | 2018 | France | Clinical isolate | 1 | CP180280-CP180286 | 75 |
| Shigella sonnei | 391324 | 2017 | UK | Clinical isolate | 1.5 | CP179997-CP180000 | NA |
| Shigella sonnei | 356538 | 2017 | UK | Clinical isolate | 2.1 | CP179993-CP179996 | 61 |
| Shigella sonnei | 830292 | 2019 | UK | Clinical isolate | 2.3 | CP179990-CP179992 | 61 |
| Shigella sonnei | 373220 | 2017 | UK | Clinical isolate | 2.12.4 | CP179986-CP179989 | 61 |
| Shigella sonnei | 590907 | 2018 | UK | Clinical isolate | 3.4.1 | CP180015-CP180021 | 61 |
| Shigella sonnei | 623218 | 2018 | UK | Clinical isolate | 3.6.1 | CP179977-CP179985 | 61 |
| Shigella sonnei | 02-1157 | 2014 | Vietnam | Clinical isolate | 3.6.1.1.1 | ERZ25074856 | 16,34 |
| Shigella sonnei | 642321 | 2018 | UK | Clinical isolate | 3.6.2 | CP179971-CP179976 | 61 |
| Shigella sonnei | 633497 | 2018 | UK | Clinical isolate | 3.7.11 | CP179964-CP179970 | 61 |
| Shigella sonnei | 598955 | 2018 | UK | Clinical isolate | 3.7.16 | CP180008-CP180014 | 61 |
| Shigella sonnei | 618335 | 2018 | UK | Clinical isolate | 3.7.28 | CP179958-CP179963 | 61 |
| Shigella sonnei | 03-0142 | 2014 | Vietnam | Clinical isolate | 3.7.29.1.4 | ERZ25074857 | 34,76 |
| Shigella sonnei | 627346 | 2018 | UK | Clinical isolate | 3.7.30.1 | CP179952-CP179957 | 61 |
| Shigella sonnei | 381259 | 2017 | UK | Clinical isolate | 3.7.30.4.1 | CP176607-CP176614 | 61 |

S. sonnei strains are referred to by genotype instead of strain ID throughout. Carb = carbenicillin, Kan = kanamycin, R = resistant. NA = not applicable. Strains with 'NA' listed under the year and country of isolation are genetically modified strains that did not undergo whole genome sequencing and were not included in genomic analyses.

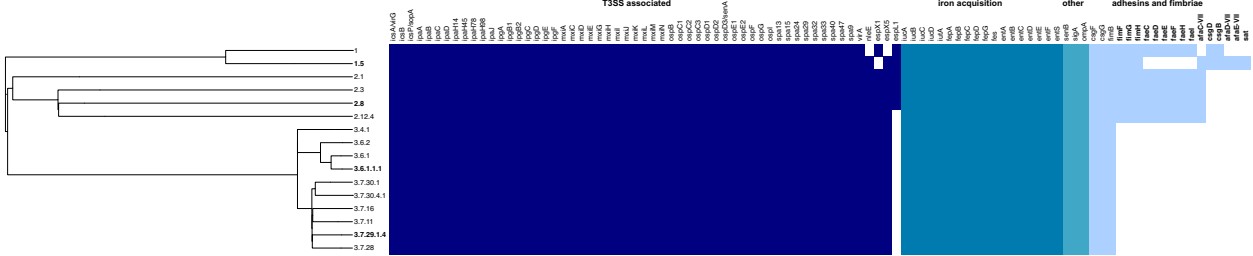

**Fig. 1 | Lineage 3 *S. sonnei* harbours fewer genes encoding for immunogenic components.** A mid-point rooted maximum likelihood phylogenetic tree with virulence associated genes identified using the virulence factor database (VFDB) plotted alongside and visualised in Phandango. Coloured squares indicate the presence of a gene, and blank squares indicate its absence. Bolded genes represent lineage-dependent differences. Bolded lineages refer to the four main lineages used for experiments. T3SS = Type Three Secretion System. Source data are provided as a Source Data file.

## pINV stability and T3SS activity do not drive differences in virulence

Considering the notorious instability of pINV in *S. sonnei*[14,45], we tested for variations in pINV stability in vitro (at 37 °C) and in vivo (infected zebrafish incubated at 32.5 °C). We found no lineage-dependent differences in pINV stability in vitro (Fig. 3a). pINV was twice as stable in vivo (Fig. 3b), but 53G was slightly more stable than Clades 3.6 and 3.7 (Fig. 3B), showing that increased virulence in zebrafish is not linked to a more stable pINV. Furthermore, these data highlight that pINV is more stable during zebrafish infection than during in vitro culturing, although this could be linked to differing replication rates in vivo.

We hypothesised that variations in T3SS activity could contribute to the virulence of Lineage 3 at 32.5 °C. Since the presence of T3SS-associated genes did not vary between lineages (Fig. 1), we tested for differences in gene expression and protein secretion. Analysis of gene expression revealed no differences in expression of pINV-encoded master regulator genes *virF* and *virB*, or of T3SS effector *ipaB* (Fig. 3c–e). Consistent with this, analysis of protein secretion revealed that total secretion of effector proteins was not greater in Lineage 3 (Fig. 3f). These results, together with genomic analysis showing limited variation in T3SS-associated gene content, support our conclusion that increased T3SS activity does not drive observed differences in virulence or epidemiological success between *S. sonnei* lineages.

## Neutrophils respond faster and in greater abundance to Lineage 3 infections

As we identified fewer immunogenic targets in Lineage 3 (Fig. 1), we hypothesised that Lineage 3 *S. sonnei* induces a different immune response in zebrafish to promote virulence, compared with 53G. Considering that *S. flexneri* is known to kill macrophages in humans and zebrafish[19,32], we measured the total number of macrophages in infected larvae over time. Macrophage numbers decreased throughout infection independently of lineage, but in most cases did not significantly differ from the uninfected control (Fig. 4a). Both *S. flexneri* and *S. sonnei* have been described to kill neutrophils[5,31,46], so we next compared the total number of neutrophils in infected larvae throughout the course of infection. In line with bacterial-induced killing, a steady decrease in neutrophil numbers was observed over time for all lineages (Fig. 4b), but no lineage-dependent differences were evident, together indicating that induction of leucocyte cell death is not responsible for the increased virulence in Lineage 3.

The zebrafish HBV typically has very few residing macrophages and no neutrophils[47] and can be used to measure recruitment of immune cells[48]. We found that leucocytes were recruited to the HBV of infected larvae as early as 3 hpi (Fig. 4c, d), with neutrophils recruited in greater abundance than macrophages, consistent with previous findings[23]. The number of macrophages recruited over time was similar for all isolates tested and did not differ by lineage (Fig. 4c). Contrastingly, the number of neutrophils recruited rose steadily until 6 hpi for all lineages; by 3 hpi more neutrophils were recruited for Clade 3.7, and by 6 hpi more neutrophils were recruited for both Lineage 3 isolates, however this difference was transitory and no longer statistically significant by 9 hpi (Fig. 4d). Together, these data highlight the dominant role of neutrophils in responding to *S. sonnei* infection of the HBV and show that neutrophils respond faster and in greater abundance to Lineage 3 infections, which likely controls survival at later timepoints.

To link these findings with inflammatory response, the expression of cytokines was quantified at 6 hpi (where differences in neutrophil recruitment were greatest). This revealed that expression of *cxcl8a*, a zebrafish homologue of interleukin-8 which plays a role in neutrophil chemotaxis[49], was significantly upregulated in Lineage 3 infected larvae (Fig. 4e). Similarly, *cxcl18b*, another chemokine involved in neutrophil recruitment[50], was upregulated (although not statistically significant) in Lineage 3 infected larvae (Fig. 4f). The expression of *il1b*, which is linked to macrophage-derived inflammation[51], was lower than other pro-inflammatory cytokines tested and no lineage dependent differences in expression were observed (Fig. 4g), in agreement with a limited role for macrophages in controlling *S. sonnei* infection[23]. Supporting a role for neutrophil-mediated inflammation in contributing to enhanced virulence of Lineage 3, we found that when the inflammatory inhibitor dexamethasone was used, differences in virulence were partly reduced (Fig. S2a, b).

## Lineage 3 *S. sonnei* isolates have an increased stress tolerance

We speculated that Lineage 3 *S. sonnei* might better withstand the bactericidal effects of the host immune response. To assess this, we measured the capacity of Lineage 3 isolates to tolerate complement-mediated killing, an important component of the innate immune system, which many host-adapted pathogens have developed strategies to overcome[52]. Following incubation with 75% complement, ~2.5-fold more CFU was recovered for Lineage 3 isolates (Fig. 5a); this difference was abolished once serum was treated at 56 °C (inactivating the complement system) (Fig. 5b). This result suggests an increased resistance against complement mediated killing, a phenotype that may contribute to increased fitness during host infection.

The O-antigen of *S. sonnei* has been shown to mediate neutrophil tolerance during zebrafish infection[5]. The O-antigen encoding regions were highly conserved between lineages; however, a comparative alignment highlighted variations in the composition of insertion sequences (ISs) upstream (Fig. S3). Considering that ISs have been implicated in driving changes in capsule expression in *Neisseria meningitidis*[53], we considered that there could be variations in outer surface layers of Lineage 3 *S. sonnei* that may drive increased stress tolerance compared to other lineages. To determine if O-antigen chain length varies between *S. sonnei* lineages, expression levels of *wzzB* (involved in O-antigen chain length modulation[54,55] (Fig. S4a, b)) were analysed by qRT-PCR. The relative mRNA expression levels of *wzzB* were similar between all isolates tested (Fig. S4c), suggesting no

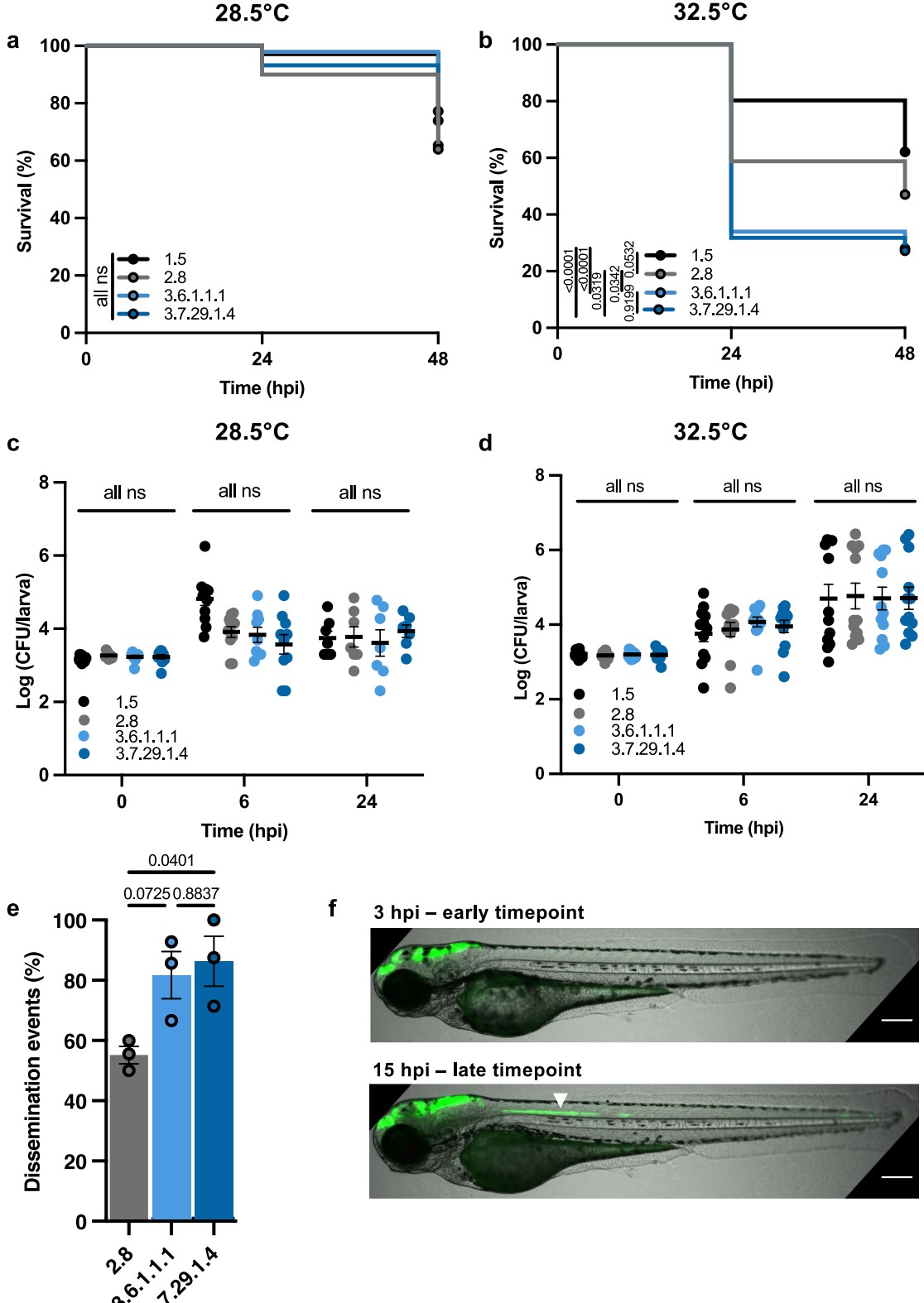

**Fig. 2 | Lineage 3 *S. sonnei* is most virulent in zebrafish. a, b** Survival curves of larvae infected with *S. sonnei* incubated at either 28.5 °C or 32.5 °C, *N* = 3 with ≥12 larvae per condition/experiment. **c, d** Log10 transformed CFU counts of infected larvae at either 0, 6 or 24 h post-infection (hpi) incubated at either 28.5 °C or 32.5 °C. *N* = 3 biological replicates with 3–4 larvae per timepoint in cases where larvae remained viable (mean ± SEM). **e** Dissemination of *S. sonnei* from the hind-brain ventricle (HBV) at 12 hpi, *N* = 3 biological replicates with 8 larvae per group (mean ± SEM). **f** Representative images depicting bacterial spread at early (3 hpi) and late (15 hpi) timepoints. Bacterial spread to the neural tube is indicated by a white arrow, scale bar = 200 µM. **Statistics:** log-rank (Mantel-Cox) test (**a**, **b**), two-way ANOVA with Tukey's correction (**c**, **d**), one-way ANOVA with Tukey's correction (**e**). ns not significant, hpi hours post infection. Source data are provided as a Source Data file.

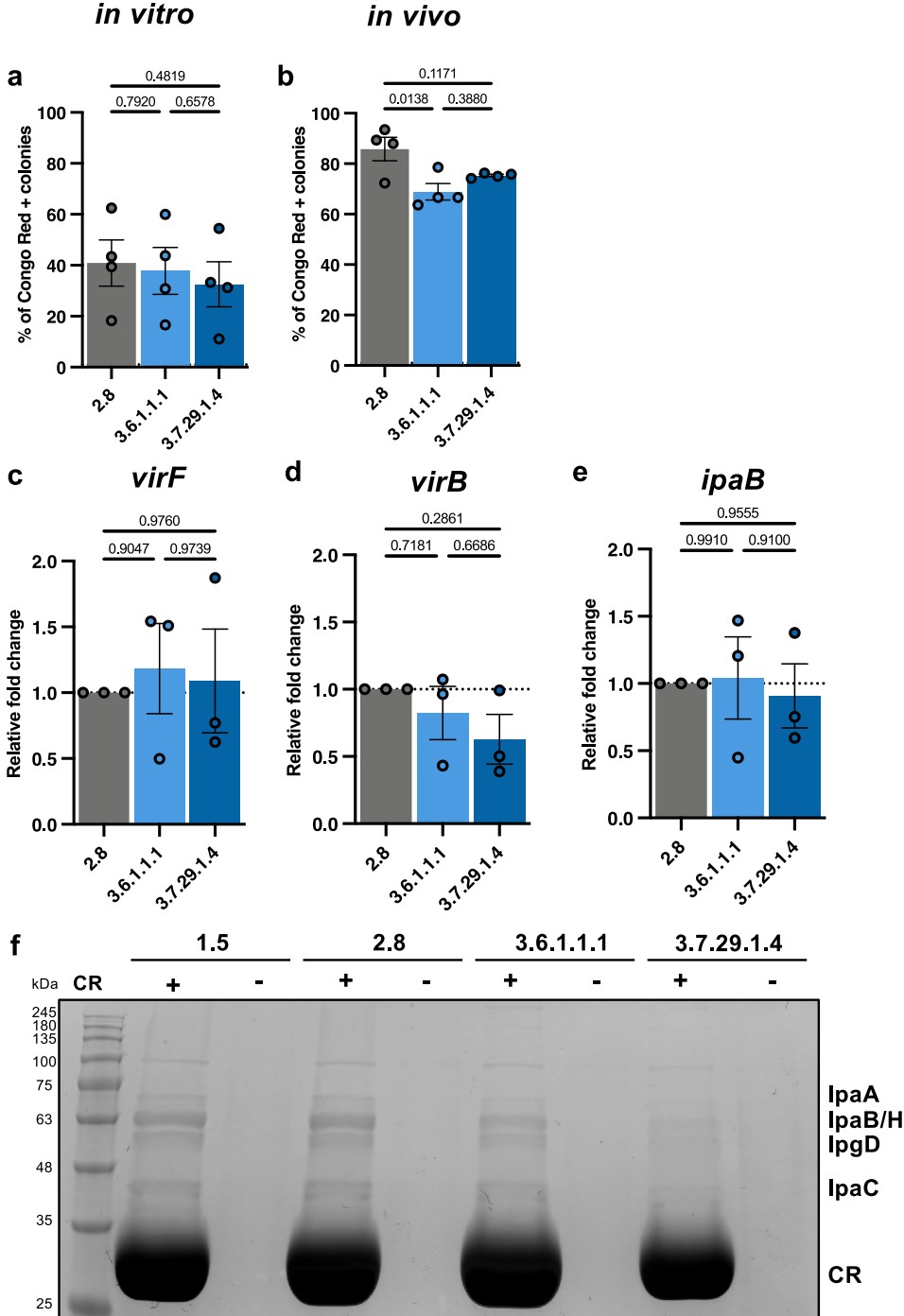

**Fig. 3 | pINV stability and T3SS activity does not drive enhanced virulence of Lineage 3 *S. sonnei*. a** Ratio of Congo red positive colonies to the total number of colonies from bacteria grown in vitro, *N* = 4 biological replicates (mean ± SEM). **b** Ratio of Congo red positive colonies to the total number of colonies from bacteria isolated from infected zebrafish *N* = 4, performed in duplicate (mean ± SEM). **c**–**e** Relative mRNA expression of T3SS genes analysed using qRT-PCR, *N* = 3 biological replicates, performed in duplicate (mean ± SEM). **f** SDS-PAGE gel stained with Coomassie blue, depicting secreted proteins in the presence or absence of Congo red (CR). **Statistics:** one-way ANOVA with Tukey's correction applied, Source data are provided as a Source Data file.

differences in O-antigen chain length. Crude LPS was extracted from bacteria and visualised, confirming that all *S. sonnei* lineages tested had a short-chain O-antigen length, with a similar number of repeating units (Fig. S4d).

Having ruled out a role for varying O-antigen chain lengths, we instead hypothesised that the G4C (Fig. S4a), which has previously been implicated in the modulation of *S. sonnei* virulence and resistance to complement-mediated killing[24], may vary in Lineage 3 isolates. To test this, we measured expression of five capsule synthesis genes which are all essential for G4C formation[56] (Fig. 5c). This revealed a consistent upregulation of gene expression in Lineage 3 isolates compared to representatives from Lineages 1 and 2, especially in Lineage 3.7 (although differences were not statistically significant for each gene independently) (Fig. 5d). Together, this establishes lineage-dependent variations in the expression of G4C, which likely go towards explaining observed differences in resistance to complement-mediated and phagosomal killing.

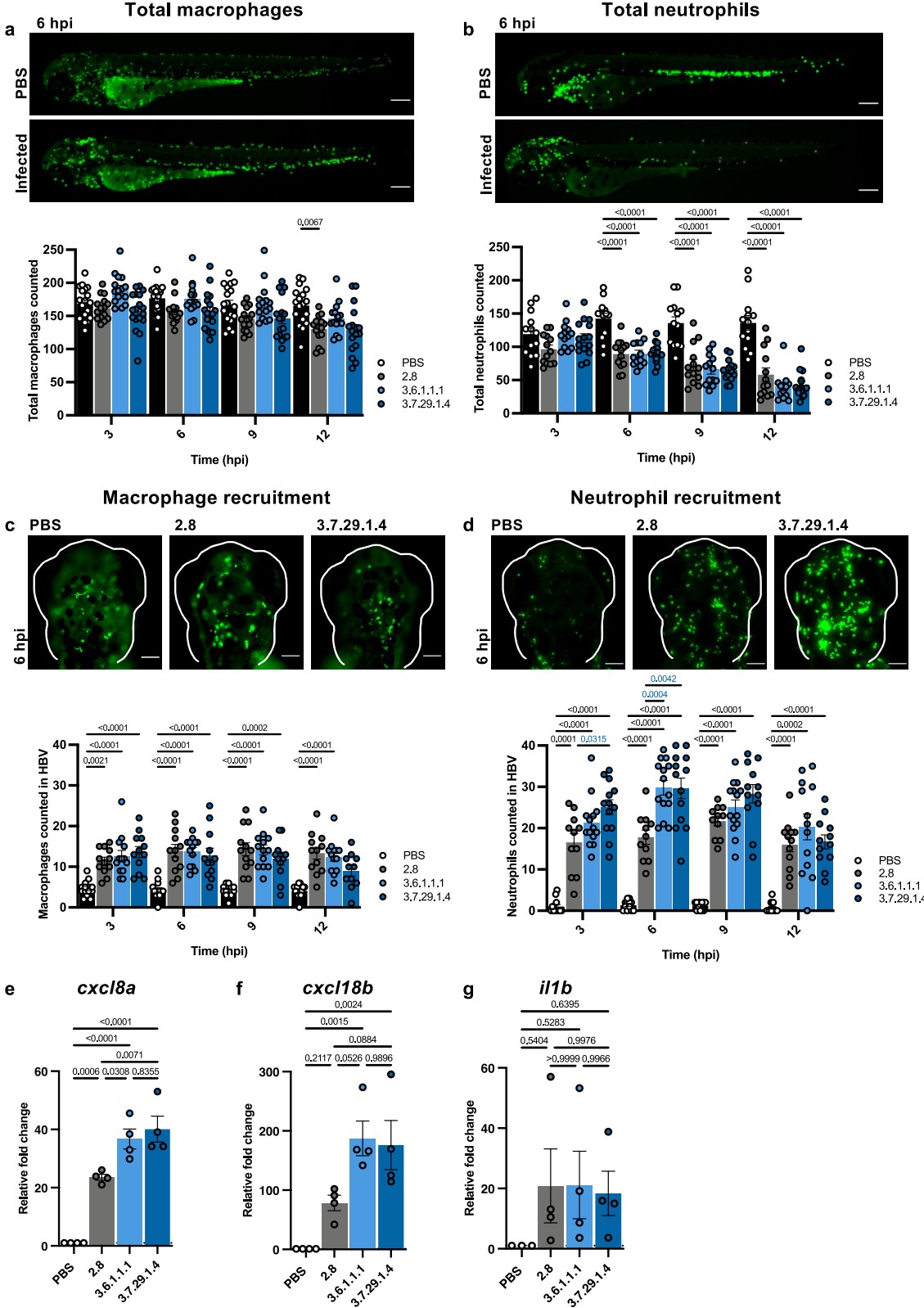

**Fig. 4 | Lineage 3 *S. sonnei* induces a stronger neutrophil response in zebrafish.** Total number of macrophages [Tg(*mpeg*1::*Gal4-FF*)[gl25]/Tg(*UAS:LIFEACT-GFP*)[mu271]] (**a**) and neutrophils [Tg(*mpx*::eGFP)[i114]] (**b**) in *S. sonnei* infected zebrafish larvae, *N* = 3 biological replicates with >4 larvae per timepoint, per replicate (>12 in total) (mean ± SEM) Scale bar = 100 μM. Number of neutrophils (**c**) and macrophages (**d**) recruited to the hindbrain ventricle (HBV) over time, *N* = 3 biological replicates with >4 larvae per timepoint, per replicate (>12 in total)(mean ± SEM) Scale bar = 200 μM. *P* values shown in blue highlight lineage-dependent differences. **e**–**g** Relative mRNA expression of pro-inflammatory cytokines from infected larvae at 6 h post-infection (hpi), *N* = 4, performed in technical duplicates (mean ± SEM). Statistics**:** Significance was tested with a two-way ANOVA with Tukey's correction ((**a**–**d**), only significance where *P* < 0.05 are displayed due to space limitations), one-way ANOVA with Tukey's correction (**e**–**g**). Source data are provided as a Source Data file.

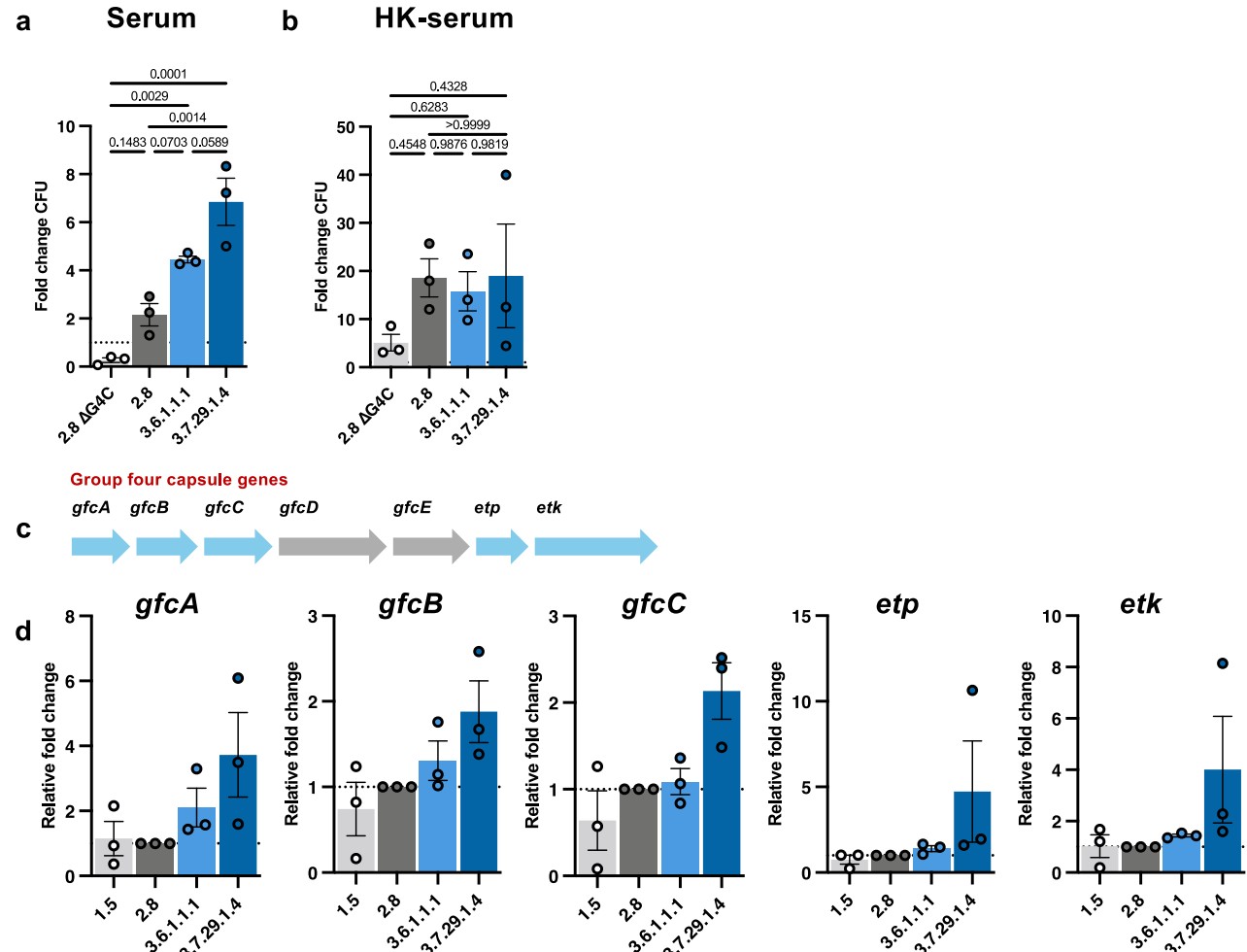

**Fig. 5 | Lineage 3 *S. sonnei* has an increased stress tolerance in vitro. a** Colony forming unit (CFU) counts of bacteria following incubation in 75% baby rabbit serum (complement), *N* = 3 biological replicates (mean ± SEM). **b** CFU counts following incubation in heat-killed (HK) serum, *N* = 3 biological replicates (mean ± SEM). **c** Schematic of the group four capsule (G4C) gene cluster. **d** Relative mRNA expression of G4C genes from bacteria analysed through qRT-PCR, *N* = 3 biological replicates, performed in technical duplicates (mean ± SEM). Statistics: One-way ANOVA, with Tukey's correction (**a**, **b**), one-way ANOVA (all results were non-significant) (**d**). Source data are provided as a Source Data file.

## Lineage 3 *S. sonnei* is more tolerant of phagosomal killing by human neutrophils

Given the nature of *Shigella* as a human-adapted pathogen, it was next of great interest to confirm if lineage-dependent differences were also observed in human cells. We first infected human epithelial cells (HeLa cells) to test for differences in bacterial invasion and intracellular replication. We found that overall rates of bacterial invasion into HeLa cells were low, consistent with what has previously been reported for *S. sonnei* 53G[23]. At 1 h 40 mins, there were no significant differences in invasion between lineages, although a trend towards reduced invasion by Lineage 3 isolates was observed (Fig. 6a). Similarly, at 3 h 40 min, no variations in bacterial replication were identified between *S. sonnei* lineages (Fig. 6b, c), agreeing with bacterial burden data from zebrafish infections. Overall, these data show that Lineage 3 does not invade HeLa cells more efficiently, and does not have a replicative advantage once intracellular, suggesting that interactions with other cell types may be driving lineage-dependent virulence.

We therefore infected primary human neutrophils and measured bacterial cytotoxicity towards neutrophils using a lactate dehydrogenase (LDH, an enzyme that is released upon cell lysis) assay. For all three lineages, ~20% LDH release (when compared to a fully lysed control sample) was observed (Fig. 6d), consistent with zebrafish results indicating no differences in leucocyte killing. Pro-inflammatory cytokine expression was also measured at 1 hpi, however, minimal expression of all tested cytokines was evident (Fig. 6E). Building on results indicating an increased resistance to stressors in vitro, we next infected primary human neutrophils with *S. sonnei* and assessed bacterial survival. The total number of viable bacteria at 1 hpi decreased in all cases (relative to the inoculum), strongly suggestive of phagosomal killing (Fig. S5). Strikingly, Lineage 3 isolates demonstrated significantly greater resistance to neutrophil-mediated killing, with ~4-fold more bacteria recovered when compared with Lineage 2.8 (Fig. 6f). These findings confirm an increased propensity of Lineage 3 to survive within neutrophils.

## Discussion

From our in silico analysis of completed whole genome sequences, we reveal that epidemiologically successful Lineage 3 *S. sonnei* harbour fewer genes encoding immunogenic components compared to less prevalent Lineages 1 and 2. Using a zebrafish infection model, we reveal that Lineage 3 is most virulent in vivo; in agreement, infection of primary human neutrophils showed enhanced tolerance to phagosomal killing. Collectively, these adaptations may help to explain the epidemiological dominance of Lineage 3. The current population of *S. sonnei* is believed to have emerged only ~500 years ago[7]. Notably, *S. sonnei* possesses the largest genome of all four *Shigella* subgroups and

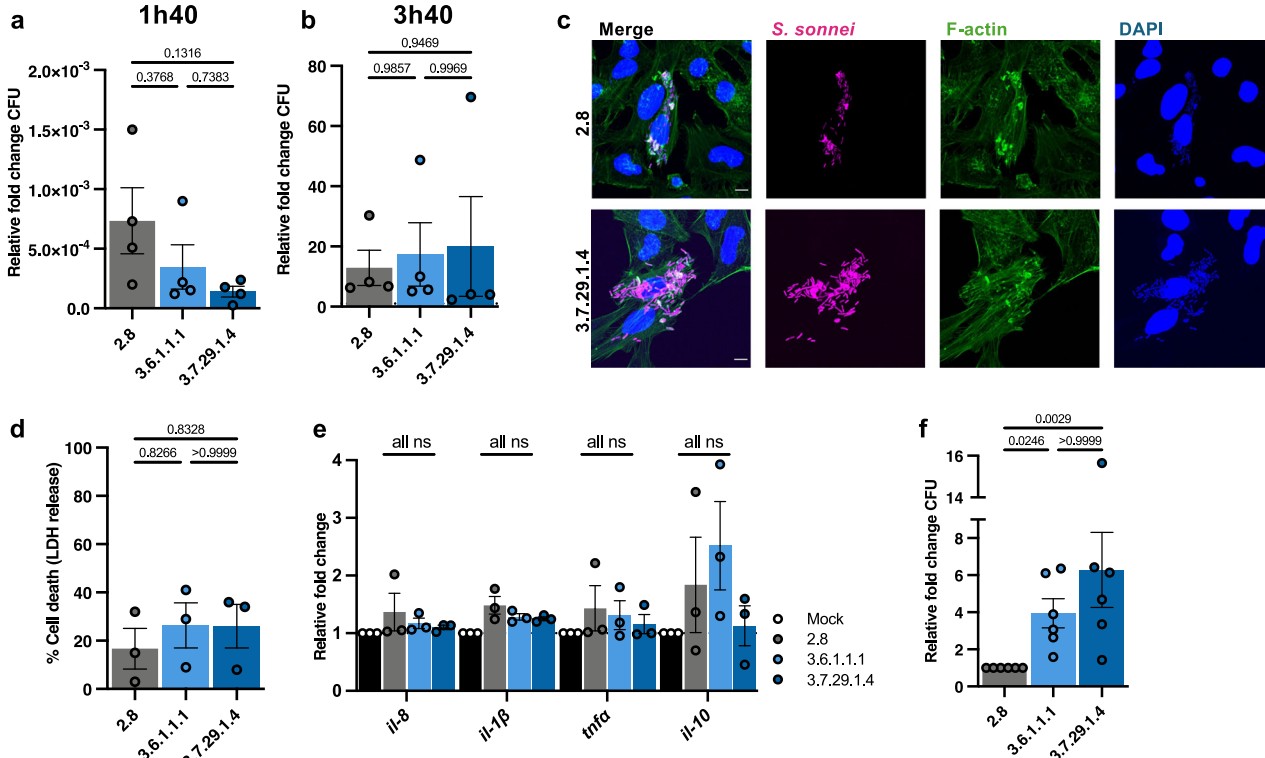

**Fig. 6 | Lineage 3 is more tolerant of neutrophil mediated killing. a** Colony forming units (CFU) obtained from infected HeLa cells at 1 h 40 min relative to the inoculum dose, N = 4 biological replicates performed in technical duplicates (mean ± SEM). **b** CFU enumerated from infected HeLa cells at 3 h 40 min, relative to the total number of intracellular bacteria at 1 h 40 min, N = 4 biological replicates, performed in technical duplicates (mean ± SEM). **c** Representative Airyscan confocal image depicting intracellular *S. sonnei* at 3 h 40 min, *S. sonnei* is shown in magenta, F-actin is shown in green, and DAPI is shown in blue; scale bar = 2 µm. **d** Cytotoxicity assay (lactate dehydrogenase (LDH) release) of infected human neutrophils normalised to an uninfected sample, N = 3 biological replicates (mean ± SEM). **e** Relative mRNA expression of *il-8, il-1β, tnfα* and *il-10* in *S. sonnei* infected neutrophils at 1 hpi, N = 3 biological replicates, performed in technical duplicates, (mean ± SEM). **f** CFU obtained from infected neutrophils at 1 h post infection (hpi) relative to the inoculum dose and normalised to Lineage 2.8, N = 6 bioloical replicates (mean ± SEM). Statistics: one-way ANOVA with Tukey's correction applied (**a, b, d–f**) ns = not significant. Source data are provided as a Source Data file.

harbours more core *E. coli* metabolic phenotypes than *S. flexneri* and *S. dysenteriae*, suggesting that it represents the least host-adapted subgroup of *Shigella*[57]. Based on our findings, together with recent work highlighting genome streamlining and IS accumulation within Lineage 3 *S. sonnei*[15], we propose that Lineage 3 *S. sonnei* are evolving towards enhanced virulence and stress tolerance, traits that are consistent with a shift towards a more host-restricted lifestyle.

A well-documented step in the pathoadaptation of *Shigella* (and other human-adapted pathogens) is the loss of immunogenic components, which can aid in immune evasion and contribute to in-host survival[39]. In agreement, our in silico analysis highlighted the overall loss of adhesins and fimbriae in Lineage 3. Our experimental data showed that Lineage 3 isolates disseminate more frequently from the infection site and invoke a greater neutrophil response. This raises the possibility that Lineage 3 *S. sonnei* can subvert or tolerate bactericidal effects of the immune system more efficiently and, as a result, may be more virulent and survive better in human neutrophils. Since we observed CFU differences in human neutrophils, and not our zebrafish model, we believe that the presence of other bactericidal components (such as macrophages and other components of the innate immune system) may mask differences in total CFUs observed from investigation of neutrophils in isolation. Furthermore, given the nature of *Shigella* as a human-restricted pathogen, such findings would next benefit from evaluation using models that more fully recapitulate clinical features of human shigellosis and more closely mimic the native host temperature, such as the NAIP-NLRC4 deficient mouse model[25], and

the controlled human infection model (CHIM)[20,26] which has been established using *S. sonnei* 53G[58].

During co-evolution with the human host, *Shigella* has evolved diverse mechanisms to efficiently infect human cells and avoid destruction[59]. In *S. sonnei*, this includes the G4C, which has been shown to confer survival advantages in vivo but compromises cellular invasion[24]. In line with this, we found that Lineage 3 isolates have an increased expression of G4C synthesis genes. These variations likely contribute to the enhanced ability of Lineage 3 to evade host immune responses (such as tolerance of complement and phagocytic destruction by neutrophils) but may also contribute to environmental survival, facilitating its spread. This pleiotropic advantage highlights an important area for further study, as understanding the balance between environmental persistence and human restriction could offer insights into the rapid clonal expansion of *S. sonnei*.

In conclusion, these data reveal that epidemiologically successful Lineage 3 *S. sonnei* isolates are more virulent in our zebrafish model than their less epidemiologically relevant counterparts. In the case of *S. sonnei*, we propose that the specific loss of immunogens and increased tolerance to stressors that arise during infection are key determinants of epidemiological success. Furthermore, we highlight the zebrafish as an important model for investigating functional variations among highly similar clinical isolates and expect that the pipeline presented here can be used to predict pathogens likely to present significant public health problems in the future.

## Methods

Animal experiments were performed according to the Animals (Scientific Procedures) Act 1986 and approved by the Home Office (Project licences: P4E664E3C and PP5900632). Human neutrophil experiments were conducted under research ethics committee reference 17/LO/1531, and written informed consent was obtained prior to conducting blood draws.

### Bacterial strains

Clinical isolates of *S. sonnei* (Table 1) were obtained in collaboration with the United Kingdom Health and Security Agency (UKHSA), Institut Pasteur, and Oxford University Clinical Research Unit (OUCRU). Strains were originally collected as part of routine public health surveillance in the UK and France[60,61], or as part of a cohort study in Vietnam[62]. Strains were received on agar slants and streaked on plates of Tryptic Soy Agar (TSA; Sigma Aldrich), supplemented with 0.01% Congo Red (CR; Sigma Aldrich) to select for pINV+ isolates. Red colonies, which indicate T3SS and pINV presence, were picked, stored in glycerol at 25% (v/v) and stored at −80 °C. Overnight cultures were prepared by inoculating 5 mL Trypticase Soy Broth (TSB; Sigma Aldrich), where necessary supplemented with 100 µg/mL carbenicillin, with a single red colony and incubating at 37 °C for ~16 h, with shaking at 200 rotations per minute (rpm).

Fluorescent bacterial strains were generated by transformation with pFPV25.1, which encodes a green fluorescent protein (GFP) reporter and resistance to carbenicillin as a selective marker. Electrocompetent cells were generated by growing bacteria to an optical density (OD) of 0.3–0.4, centrifugation at 4 °C and washing with 10% (v/v) ice-cold glycerol. 100 ng of DNA was added, and the suspension was electroporated using the Ec2 setting on a Bio-Rad Micropulser. 1 mL of TSB was immediately added, and bacteria were left to recover at 37 °C for 2 h. Following recovery, the suspension was plated on TSA supplemented with 50 µg/mL carbenicillin to select for positive colonies.

### In silico analysis

Complete *S. sonnei* genomes (accessions available in Table 1, sequencing and assembly described previously[15]) were screened for virulence using ABRicate (v.1.0.1)[63] with the VFDB option selected (accessed on 18/08/2024)[64]. MAFFT (v.7.526)[65] was used to perform a core genome alignment with default settings. The resulting alignment was used as an input to FastTree (v.2.1.1)[66], which was run using the generalised time-reversible model to produce a maximum likelihood phylogenetic tree. Visualisation was performed using Phandango (accessed on 19/12/2024)[67] and InkScape (v.1.3.2). To generate gene cluster comparison figures, relevant gene clusters were extracted and aligned using Clinker (v.0.0.29)[68]. Fimbrial operons were manually inspected and compared using Mauve (v.2.4.0)[69].

### Zebrafish infection

All zebrafish experiments were performed on larvae up to 5 days post fertilisation (dpf). Zebrafish embryos were obtained from naturally spawning larvae and incubated at 28.5 °C in 0.5 x E2 medium (15 mM NaCl, 1 mM MgSO4, 500 µM KCl, 150 µM KH2PO4, 50 µM Na2HPO4, 0.3 µg/ml methylene blue).

20 mL of TSB was inoculated with 400 µL overnight culture and grown to mid-exponential phase. Bacteria was harvested by centrifugation (4000 x *g*, 5 min), washed in 1 mL phosphate-buffered saline (PBS; Sigma Aldrich) to remove residual media and pelleted again (1 min, 6000 x *g*). The desired inoculum concentration was achieved by measuring the OD of bacteria and correction to the desired OD. Injection inoculum was prepared by resuspension of bacteria in inoculum buffer (2% polyvinyl-pyrrolidone (PVP; Sigma Aldrich), PBS and 0.5% phenol red (Sigma Aldrich)) to a final volume of 100 µL. Control groups of larvae were injected with inoculum buffer.

### Dexamethasone treatment

Dexamethasone (Sigma Aldrich) was resuspended in dimethyl sulfoxide (DMSO, Sigma Aldrich) at a concentration of 25 mg/mL. Injections were performed as described above, and recovered larvae were split into two groups, treated and control. The treated group were incubated in E2 medium containing 50 µg/mL dexamethasone, and the control group were incubated in E2 medium with the same concentration of DMSO added, as reported previously[70].

### Survival assays and bacterial burden

Larvae were maintained in groups of two or three in 24-well plates at 28.5 °C or 32.5 °C and visualised using a light stereomicroscope to check survival. The precise inoculum was determined retrospectively by the mechanical disruption of a single larva in 200 µL 0.4% Triton-X-100 (Sigma Aldrich) at 0 hpi, and at 6 and 24 hpi for bacterial burden quantifications. For each time point, four different larvae were selected at random as representatives for the infected population. Larvae homogenates were serially diluted in PBS, plated on TSA plates supplemented with CR and colonies were counted manually following overnight incubation at 37 °C.

### Eukaryotic cell work

HeLa cells (Human ATCC CCL-2) were used for eukaryotic cell infections. Cells were cultured in Dulbecco's Modified Eagle Medium (DMEM, Sigma Aldrich) supplemented with 10% foetal-bovine serum (FBS, ThermoFisher Scientific) and incubated in an incubator supplied with 5% $CO_2$ at 37 °C. Cells were seeded at a density of $1.5 \times 10^5$ in a 6-well plate (VWR) 48 h before infection. Bacterial cultures were grown to mid-exponential phase and then diluted in DMEM (not supplemented with FBS) to reach a multiplicity of infection (MOI) of 100:1. 1 mL of bacterial suspension was added to each well, and cells were centrifuged at 500 x *g* for 10 min at room temperature. Cells were then incubated for 30 min with 5% $CO_2$ at 37 °C. Bacterial suspension was removed from the cells, and the cells were washed three times with PBS. Washed cells were treated with 2000 µL of 50 mg/mL gentamycin (Sigma Aldrich) in DMEM + FBS for 1 or 3 h for invasion and replication timepoints, respectively. Following treatment, cells were lysed with 0.1% Triton-X-100 (Sigma Aldrich) in PBS for 5 min at 37 °C. Cell lysates were then serially diluted and plated on TSA, plates were incubated overnight, and CFU counts were determined the following morning.

For imaging, cells previously seeded and infected on coverslips were washed 3 times with PBS and then fixed using 4% (v/v) paraformaldehyde (PFA) in PBS for 15 min at room temperature. Fixed cells were washed 3 times in PBS and permeabilised in blocking (0.1% Triton X-100 and 1% bovine serum albumin (BSA) in PBS). Cells were stained with Hoechst 33342 (1:500, #H3570, ThermoFisher Scientific) and AlexaFluor-647 conjugated Phalloidin (1:500, #10656353, ThermoFisher Scientific) in blocking solution at room temperature in a humid chamber for 1 h. Coverslips were washed 9 times in PBS and then mounted onto glass slides using ProLong Gold antifade reagent with DAPI (#P36935, Thermofisher) mounting media. Fluorescence microscopy was performed using a ZEISS CellDiscover 7 (CD7) microscope using a ZEISS Plan-APOCHROMAT 20× / 0.95 Autocorr Objective coupled to a 0.5x tubelens. Z-stack confocal images of 32 slices over 5 µM were acquired. Confocal images were processed using Airyscan processing (Weinerfilter) using "Auto Filter" and "3D Processing" options.

### Human neutrophil infections

Blood was drawn from a healthy donor, using EDTA as an anticoagulant (BD Vacutainer, Becton Dickinson). Neutrophils were isolated using Polymorphprep (Serumwerk Bernburg) solution as per manufacturers guidelines. Briefly, 5–7 mL of blood was slowly layered over an equal quantity of Polymorphprep solution, and the layered solution was centrifuged for 30 min at 500 x *g* with decreased acceleration and no

**Table 2 | Primers used to measure *S. sonnei* gene expression via qRT-PCR**

| Primer name | Sequence (5'-3') | Source |
|---|---|---|
| *rrsA_FW* | AACGTCAATGAGCAAAGGTATTAA | 77 |
| *rrsA_RV* | GAACTTCAAGATCTGCTCCTGC | 77 |
| *etp_FW* | CTCAATCCTGGTGGTTTGTACCG | 78 |
| *etp_RV* | GACTCCATTGCCAGAATCAGATC | 78 |
| *etk_FW* | CAGGCAGCACTCAGGAAAATGAG | 78 |
| *etk_RV* | GATTGCAGCAGTTGGATCTCCG | 78 |
| *gfcA_FW* | CTTTCTGCAATCCTTATGGCCTTC | 78 |
| *gfcA_RV* | GACGTGGTGGTTGAGGTGTTATG | 78 |
| *gfcB_FW* | CACTATTTCTTGCGGGATGTACG | 78 |
| *gfcB_RV* | GTCCATTGTGGGTAACCAGCATG | 78 |
| *gfcC_FW* | CAGTCGTATTTCATTGCCAGCG | 78 |
| *gfcC_RV* | GACATGTTGGTAGTCTTTAAGCGC | 78 |
| *virF_FW* | AAAGGTGTTCAATGACGGTTAGC | 79 |
| *virF_RV* | CAATTTGCCCTTCATCGATAGTC | 79 |
| *virB_FW* | GGAAGGCCAAAAGAAAGAGTTTACA | 79 |
| *virB_RV* | GAGGAATCTTGGCTTTGATAAAGG | 79 |
| *ipaB_Fw* | CTGCATTTTCAAACACAGC | 77 |
| *ipaB_Rv* | GAGTAACACTGGCAAGTC | 77 |
| *wzzB_Fw* | GCGATAACATTCAGGCGCAA | This study |
| *wzzB_RV* | CCCCTGGTAATGCACCAAGA | This study |

brakes on deceleration. The neutrophil layer was collected and washed with 10 mL 0.5% (v/v) PBS. Where red blood cells (RBC) were present, a lysis step was incorporated, where 3 mL of 1X RBC lysis buffer (Invitrogen) was added per 5 mL of blood collected and incubated for 10 min. Following RBC lysis, neutrophils were washed twice as previously described and finally resuspended in neutrophil medium (RPMI 1640 Medium, GlutaMAX™ Supplement, Sigma Aldrich). Neutrophils were counted using Trypan Blue Staining and resuspended to the desired concentration.

Infections were performed in a 48-well plate (VWR), with $10^5$ neutrophils added per well. *S. sonnei* was cultured as previously described, and $10^3$ bacteria were added per well, with experiments performed in technical duplicates. 20 μL of the initial inoculum was plated to determine precise bacterial input. The 48-well plate was incubated for 1 h at 37 °C with 5% $CO_2$. For CFU determination, 7.5 μL of 0.1% Triton-X was added, and neutrophils were placed on ice to lyse as previously reported[5]; to prevent excessive neutrophil cell death, cells were not centrifuged prior to lysis. A 10-fold serial dilution was next performed, and 20 μL of lysate at each dilution was plated. CFUs were counted the following morning, and bacterial survival was determined by normalising CFU at 1 hpi to the initial inoculum.

## Bacterial RNA extraction and qRT-PCR
Bacterial cultures were grown to mid-exponential phase, as described above. An amount of culture corresponding to $1 \times 10^9$ bacterial cells was pelleted by centrifugation at $5000 \times g$ at 4 °C for 10 min, the supernatant was removed, and the pellet was kept at −80 °C overnight to aid with cell lysis. RNA was then extracted using the Monarch Total RNA Miniprep Kit (New England Biolabs) as per the manufacturer's instructions. RNA concentration was measured using a DeNovix DS-11 spectrophotomer.1000 ng of RNA was then converted to cDNA using a QuantiTect reverse transcription kit (Qiagen). Template cDNA was subjected to quantitative reverse transcription PCR (qRT-PCR) using a 7500 Fast Real-Time PCR System machine and SYBR green master mix (Applied Biosystems), with samples run in technical duplicates. Primers generated for this study were designed using the NCBI Primer Design Tool (https://www.ncbi.nlm.nih.gov/tools/primer-blast/), and

all primers used can be found in Table 2, *rrsA* was used as a housekeeping gene, and the delta-delta Ct method was used to quantify gene expression.

## Zebrafish RNA extraction and qRT-PCR
For each condition, ~15 embryos were pooled and frozen overnight at −80 °C. RNA was then extracted using the RNAeasy Minikit (Qiagen), converted to cDNA and subjected to qRT-PCR as previously described. Primers used for zebrafish qRT-PCR can be found in Table 3. Zebrafish gene *eef1a1a* was used as a housekeeping gene, and the delta-delta Ct method was used to quantify changes in gene expression.

## Zebrafish microscopy
For imaging, larvae were anaesthetised in 1X tricaine (Sigma Aldrich), placed into a 96-well plate (Perkin Elmer) and embedded in 1 drop of 1% low-melting point agarose (w/v, Thermo Scientific), and wells were topped up with tricaine for the duration of the imaging process. For leucocyte counts, larvae were imaged using a Leica M205FA microscope and counts were performed manually.

A Zeiss Celldiscoverer 7 (CD7) microscope was used for acquiring representative images. For whole embryo imaging, larvae were placed laterally, and a 5 x /0.35 plan-apochromat objective with a 0.5 x tube lens was used to capture widefield z-stacks of 13 slices of 18.46 μM each. For HBV imaging, larvae were placed head-down and widefield imaging was performed using the same objective, but with a 2 x tube lens, to capture z-stacks across the HBV.

## Bacterial resistance assay
One fresh red colony was inoculated into 50 μL PBS. Lyophilised baby rabbit complement (Bio-Rad, C12CA) was resuspended in 2 mL ice-cold water, and 150 μL was added to the bacterial suspension. 10 μL was taken for serial dilution and plating to determine the initial inoculum. The bacterial-rabbit complement mixture was incubated at 37 °C, shaking at 200 rpm for 4 h before another 10 μL was taken, serially diluted and plated. CR+ colonies were counted the following morning, and fold change CFU was calculated by normalising colonies counted at 4 h to the initial inoculum. As a control, baby rabbit complement was heat killed at 56 °C for 30 min to inactivate the complement system, as previously reported[71,72].

## Protein secretion assay
To analyse the secretion of T3SS effector proteins, bacterial cultures were grown until OD 0.2–0.3 and CR was added at a concentration of 200 μg/mL. Cultures were then grown to maximum OD (2–3). Bacterial cells were pelleted by centrifugation at 10,000 x $g$, at 4 °C for 10 min. The supernatant was then removed and filtered using 0.2 μm filters to remove cell debris. A volume of filtered supernatant corresponding to OD 2 was taken and trichloroacetic acid (Sigma Aldrich) was added at 10% (v/v) concentration, this mixture was then incubated at −20 °C overnight. The following day, samples were pelleted at 4 °C at maximum speed and washed with 1 mL ice-cold acetone (Sigma Aldrich) 3 times to remove any residual acid. Samples were resuspended in 25 μL 1X Laemmli buffer (10 mM Tris-HCl, pH 6.8, 2% sodium dodecyl 63 sulphate [SDS], 10% glycerol, 5% β-mercaptoethanol, 0.01% bromophenol blue) prior to SDS-PAGE.

## SDS PAGE
Protein samples were boiled at 100 °C, 17.5 μL of sample was loaded onto 12% SDS polyacrylamide gels and gels were run at 100 V in 1X Tris-glycine-SDS running buffer in a Bio-Rad Protean Tetra cell. For secretion experiments, gels were rinsed with distilled water, then stained using Coomassie Brilliant Blue R-250 (Bio-Rad) overnight, before destaining in a solution of methanol, acetic acid, and water (30%, 5% and 65% (v/v) respectively).

**Table 3 | Primers used to measure zebrafish gene expression via qRT-PCR**

| Primer name | Sequence (5'-3') | Source |
|---|---|---|
| *eef1a1a*FW | AAGCTTGAAGACAACCCCAAGAGC | 80 |
| *eef1a1a*RV | ACTCCTTTAATCACTCCCACCGCA | 80 |
| *cxcl8a*FW | TGTGTTATTGTTTTCCTGGCATTTC | 81 |
| *cxcl8a*RV | GCGACAGCGTGGATCTACAG | 81 |
| *cxcl18b*FW | TCTTCTGCTGCTGCTTGCGGT | 50 |
| *cxcl18b*RV | GGTGTCCCTGCGAGCACGAT | 50 |
| *il1b*FW | GAACAGAATGAAGCACATCAAACC | 81 |
| *il1b*RV | ACGGCACTGAATCCACCAC | 81 |
| *il10*FW | CATAACATAAACAGTCCCTATG | 82 |
| *il10*RV | GTACCTCTTGCATTTCACCA | 82 |
| *tnfa*FW | AGACCTTAGACTGGAGAGATGAC | 81 |
| *tnfa*RV | CAAAGACACCTGGCTGTAGAC | 81 |

## LPS extraction

Crude LPS extraction was carried out as described previously[73]. Bacteria were grown overnight and then sub-cultured to reach mid-exponential phase as described above. Samples were normalised by OD to ensure an equal density of bacteria in each sample before pelleting at $10,000 \times g$ for 10 min at 4 °C. Pelleted bacteria were resuspended in 200 µL 1X Laemmli buffer and boiled for 15 min. 5 µL DNAse I and RNAse (10 mg/mL) was added, and the mixture was incubated at 37 °C for 30 min. 10 µL of Proteinase K (10 mg/mL) was then added, and the mixture was incubated at 59 °C for a further 3 h. Following this, 200 µL of ice-cold Tris-saturated phenol was added to each sample. Samples were then heated to 65 °C for 15 min, with occasional vortexing and once cool, 1 mL petroleum ether was added. Samples were centrifuged at $14,000 \times g$ for 10 min; the aqueous layer was isolated and added to 150 µL Laemmli buffer. 10 µL was added to a 12% SDS polyacrylamide gel and run as described above.

LPS was then visualised using a modified silver stain, which oxidises LPS, allowing for better visualisation[74]. Briefly, following SDS-PAGE, gels were rinsed with water and then incubated with a fixing solution (40% ethanol, 5% acetic acid) overnight. Fixing solution was then replaced with an oxidising solution (0.7% periodic acid, 40% ethanol, 5% acetic acid) for 20 min. Following the oxidation step, the gel was washed three times in distilled water for 10 min and was then stained using the Pierce Silver Staining Kit according to the manufacturer's directions. All gels were visualised using a ChemiDoc Touch Gel Imaging System.

## Reporting summary

Further information on research design is available in the Nature Portfolio Reporting Summary linked to this article.

## Data availability

The sequence data analysed in this study are publicly available via NCBI under the accessions listed in Table 1. Data generated in this study are provided in the source data file. Biological materials are available upon request to the corresponding author. Source data are provided with this paper.

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

## Acknowledgements

We thank Mostowy lab members for helpful discussions and the LSHTM Biological Services Facility for the work and care of zebrafish stocks. Funding: SLM was supported by a Biotechnology and Biological Sciences Research Council LIDo Ph.D. studentship (BB/T008709/1). This research in the SM laboratory was supported by a Wellcome Trust Senior Research Fellowship (206444/Z/17/Z), European Research Council Consolidator Grant (772853 - ENTRAPMENT), MRC Impact Accelerator Account award (IAA21127) and Wellcome Discovery Award (226644/Z/22/Z).

## Author contributions

S.L.M. and S.M. wrote the manuscript; all other authors provided feedback on the manuscript. S.L.M., K.H. and S.M. interpreted the data. S.L.M. performed most experiments, including all genomics analyses and zebrafish experiments. D.S. performed neutrophil extractions. A.T.L.J. assisted with protein secretion assays. V.T. performed preliminary experiments. H.P., K.W., M.V., X.M.M., A.C., D.C. and A.P. assisted with the revision process. C.J., S.B. and K.S.B. provided bacterial strains. V.S.S., K.H. and S.M. supervised the study. K.H. and S.M. conceived the study.

## Competing interests

We declare no competing interests.

## Additional information

[1]Department of Infection Biology, London School of Hygiene and Tropical Medicine, London, United Kingdom. [2]Section of Paediatric Infectious Disease, Department of Infectious Disease, Faculty of Medicine, Imperial College London, London, UK. [3]Centre for Bacterial Resistance Biology, Department of Life Sciences, Imperial College London, London, UK. [4]Gastrointestinal Bacterial Reference Unit, UK Health Security Agency, London, United Kingdom. [5]A*STAR Infectious Disease Labs, Biopolis, Singapore, Singapore. [6]Department of Genetics, University of Cambridge, Cambridge, UK. [7]Wellcome-Wolfson Institute for Experimental Medicine (WWIEM), School of Medicine, Dentistry and Biomedical Sciences, Queen's University Belfast, Belfast, UK. [8]Department of Molecular Biology, Umea University, SE-901 87 Umea, Sweden. [9]Centre for Paediatrics and Child Health, Faculty of Medicine, Imperial College London, London, UK. [10]Section of Virology, Department of Infectious Disease, Faculty of Medicine, Imperial College London, London, UK. [11]Department of Infectious Diseases, School of Translational Medicine, Monash University, Melbourne, Australia. [12]Present address: Department of Infectious Diseases, School of Immunology and Microbial Sciences, King's College London, London, UK. ✉e-mail: serge.mostowy@lshtm.ac.uk

