## [Transparent Peer Review file · Nature Communications]

Enhanced virulence and stress tolerance are signatures of epidemiologically successful *Shigella sonnei*

Corresponding Author: Professor Serge Mostowy

Version 0:

Reviewer comments:

Reviewer #1

(Remarks to the Author)

The manuscript from Miles et al. addresses virulence phenotypes of *Shigella sonnei* lineages that are well distributed in humans, lineage 3 being the most frequent among the 5 *S. sonnei* lineages. The study initially focuses on lineages 1-3, before a detailed analysis on two strains of lineage 3 that are compared to the well-known 53G strain (of lineage 2). *S. sonnei* strains of lineage 3 harbour fewer genes encoding adhesion and fimbriae. The virulence of strains was assessed using zebrafish embryo as infection model at an unusual temperature (32,5°C), which allowed to discriminate lineage 3 from lineage 2. Lineage 3 strains are more virulent in zebrafish, where they exhibit a greater neutrophil response. In human neutrophils, lineage 3 strains exhibit greater tolerance to phagosomal killing. In addition, increased stress tolerance (to acid stress and complement) was observed for lineage 3 in vitro, as well as increased expression of group 4 capsule (G4C) genes. Authors conclude that cumulatively, these features may explain the epidemiological success of lineage 3 of *S. sonnei*.

A better characterization of *S. sonnei* pathogenicity/epidemiology is of major interest.

Whereas the study provides an appealing new picture on *S. sonnei* lineage 3, few additional experiments are required to strengthen the findings and the conclusions. Several interesting observations are rather mild and lack experimental demonstration for causal links, which limits the impact of the conclusions (as proposed L265-273, L276-281, L285-292).

Major points

1) *S. sonnei* strains of lineage 3 harbour fewer genes encoding immunogenic components (adhesin and fimbriae) and express slightly more G4C. However, no experimental evidence supports that these feature have a causal link with higher virulence in zebrafish / enhanced stress tolerance / increased resistance to human neutrophils. Moreover, the link between stress tolerance and virulence phenotypes is not addressed.

Additional experiments should be considered to address these points. For example, since a *S. sonnei* 53G Δ tag4c is used in the study, a similar mutant could be constructed in a lineage 3 strain ? It would allow to address the contribution of G4C in some phenotypes.

To assess the impact of higher tolerance to acid stress on lineage 3 strains in neutrophils / virulence phenotypes, authors should treat human neutrophils and zebrafish with an inhibitor of acidification (as bafilomycin). If the inhibition of acidification suppresses the difference between lineage 3 and lineage 2 strains (CFUs for neutrophils and embryo survival for zebrafish), it would support a link between acid stress tolerance of lineage 3 and increased tolerance to phagosomal killing / increased virulence.

2) CFUs from zebrafish. The profiles of time 0 are identical in panels C and D of Fig 1: if they represent inoculum, and not initial CFUs from infected fish, they should be removed from the graph.

Whereas CFUs do not differ between strains at 24 hpi at 28,5°C, the authors make the same conclusion at 32,5°C, whereas the data suggest that there may be different trends (Fig 2D). However, the number of embryos tested at 24 hpi in the condition 32,5°C is too low (5 embryos), probably due to the high mortality rate. More embryos should be infected to have at least 12-15 embryos alive at 24 hpi for CFUs quantification. This is required to conclude if there is a difference or not in CFUs between the 4 strains at 24 hpi.

In addition, authors speculate that lineage 3 of *S. sonnei* better withstand the bactericidal effect of the host immune

response: how do they confront this hypothesis with the conclusion that no difference in CFUs are observed (Fig. 2D) ? An increased CFU number in zebrafish should be expected at some point with lineage 3 compared to lineage 2.

3) Stress tolerance: the differences between lineage 2 and 3 are rather mild (not significant for 3.6 isolate in Fig 5A) and no statistical analysis in Fig 5D. These experiments are simple and should be strengthened by extending the analysis (growth at pH 5 should be evaluated on a longer time period, at least 6-7 hrs) and conducting experiments on all strains tested for virulence in Fig 2 and Fig S1 to have a broader / clearer picture of the impact of the results and justify the title.

4) Expression of G4C genes (Fig. 5F and 5H): the effect on *etk* and *etp* expression is mild and not significantly different for *etp*. Other genes of the operon (as *gfcA*, which is also implicated in the formation of the capsule) could be tested as well. In addition, the study could be extended to few other strains (at least 1 from each lineage ?) to strengthen the results and drive a conclusion.

The conclusion L233-235: "Together, this establishes lineage-dependent variations in the expression of G4C, which likely explain observed differences in stress tolerance" is not supported by the data since a causal relation is not demonstrated (this would imply mutating G4C encoding genes in a lineage 3 strain for example).

5) Phenotypes in human neutrophils (Fig 6): the two isolates of lineage 3 showed an increased propensity to survive within neutrophils. If this is also the case in the zebrafish model, again one would expect higher CFUs for lineage 3 strains at some point. In addition, authors could take advantage of the Tg(*mpx::eGFP*) fish line (Fig. 4D) to visualize the interaction between neutrophils and red fluorescent *S. sonnei* strains (as used in their previous publication, reference 33) at 6 hpi and 24 hpi.

Minor points

1) The review "Pathogenicity and virulence of *Shigella sonnei*: A highly drug-resistant pathogen of increasing prevalence. Matanza XM, Clements A. *Virulence*. 2023. doi: 10.1080/21505594.2023.2280838." should be cited

2) Survival curves: the higher virulence of lineage 3 is based on the analysis of 2 representative strains (Fig 2). Additional lineage 3 strains are tested in Fig S1 but Fig S1D is difficult to read due to a high number of strains (at time 48 hpi only 6 points out of 7 are visible). It seems that some strains of lineage 3 have profiles close to lineage 2 (strain 3.4 in panel C and 3.7.30.4.1 in panel D ?). It could be of interest to see how these two strains behave for other phenotypes ?

3) Fig 4D: The authors should insist on the fact that the higher recruitment of neutrophils with lineage 3 is transitory (visible at 6hrs but not at 9h)

4) Zebrafish infection experiments are known to show some variability, as seen with the survival curves of Fig 2B and S2A. In figure S2A, the difference between lineage 3 and 2 is clear only after 24 hpi (which is not the case in figure 2B). Similarly, the differential effect of dexamethasone is only visible between 24 and 48 hpi (before 24 hpi, a similar reduced virulence is observed for lineages 2 and 3). How do the authors explain these findings considering the timing of the neutrophil response (6 hpi) ?

5) Strain 53G (2.8) was previously reported by the same team to establish persistent infection in zebrafish at 28°C. Do the virulent strains from cluster 3 also establish a persistent infection ? If not tested, this could be at least discussed ?

L110-111: please clarify

Fig 1: the name of the genes that show differences among lineage 3 strains should be enlarged (difficult to read). The 4 strains mainly tested in the study should be easily identified (bold ?)

L147: significant only for one of the two strains

L156-157: the replication rate could make a difference for pINV stability in vitro / in vivo, thus this does not necessarily support that pINV is a key driver of *S. sonnei* in vivo

L201 (or Mat & Met): a reference with the use of DMX in zebrafish should be added

L229: write "group 4 capsule" ?

L283: a specific reference for CHIM should be provided (rather than 2 reviews)

L281-284: In addition to the CHIM, authors could also mention that it would be of interest to translate the virulence profiles obtained in zebrafish in the NAIP-NLRC4 deficient mouse model

Reviewer #2

(Remarks to the Author)

Summary: The goal of the manuscript was to compare different lineages of *Shigella sonnei* to identify factors that contribute to the infection dominance and epidemiological success of certain *S. sonnei* isolates. The study uses isolates from three

separate lineages of *S. sonnei* and includes lab strains *S. sonnei* 53G and *S. flexneri* strain M90T for comparisons. The authors utilized comparative genomics to identify virulence genes harbored by the isolates and examined infection and dissemination in a zebrafish model. Subsequently, analyses were performed to further examine how certain *S. sonnei* isolates could effectively infect the model; and thus, examined stability of the virulence plasmid with secretion of virulence proteins, measured macrophage and neutrophil responses in the zebrafish, and assessed tolerance to stresses and phagosomal killing. Based on the data, the authors conclude that Lineage 3 isolates are more virulent and are more tolerant of stresses, which is related to upregulation of group four capsule synthesis genes. Combined with the epidemiological data of Lineage 3 isolates, the increased virulence and stress tolerance data provide reasons for the global spread of this lineage. Overall, the study enhances our understanding of *S. sonnei* infection and demonstrates the usefulness of incorporating the zebrafish model into *Shigella* studies.

Overall, the authors provide a well-written manuscript with reproducible data, utilize complementary methodology to validate their findings, and demonstrate appropriate number of replicate experiments and statistical analyses for each experiment. The data are convincing, but review of the manuscript identified some concerns that should be addressed by the authors. These concerns are outlined below.

1. For the genome comparisons, some clarifications would be helpful or are needed:

a. While M90T is a common and appropriate *S. flexneri* strain for comparison purposes, other *S. flexneri* isolates like serotype 2a strains harbor additional virulence genes and pathogenicity islands that are highly immunogenic or modulate the host immune response. These genes include, but are not limited to, *pic*, *sepA* and *sigA*. It would be useful to know if any of the *S. sonnei* isolates have acquired these genes. Further, consideration of these other virulence genes when comparing the isolates may help to determine potential roles for these virulence genes in *S. sonnei* given the data the authors obtained.

b. Regarding the adherence genes, the authors list genes that are absent in *S. sonnei*. For example, *fimBHGF* are absent in lineage 3 (as noted on line 119). These genes are relatively minor with regards to type 1 fimbriae expression as *fimB* encodes a recombinase for phase variable switching and *fimHGF* encode minor subunits of the fimbrial structure. It is unclear if critical genes such as *fimA* encoding the major subunit, *fimC* encoding the chaperone, and *fimD* encoding the pore/usher are present. Likewise, for *curli* and based on the wording of the manuscript, it is unclear if the major subunit *csgA* or the pore gene *csgG* are present. If these genes have been maintained, it is possible the proteins are expressed. As demonstrated in the citation *mSphere* (2019) 4(6):e00751-19, doi: 10.1128/mSphere.00751-19, *S. flexneri* expresses adherence factors despite missing or non-functional genes.

2. For the zebrafish model, the authors do not provide uninfected controls, especially for analyses performed at 32.5 C. Since zebrafish are typically cultured between 26-28.5 C, it is possible that the zebrafish were affected or even in some instances killed at 32.5 C. Zebrafish killing due to temperature will affect interpretation of the results. Finally, it would be helpful if the authors provided more context for temperature-regulated virulence in *Shigella*. A helpful citation is *Front Mol Biosci.* (2016) 3:61. doi: 10.3389/fmolb.2016.00061 that explains that virulence activation does start above 32 C.

3. For the Congo red secretion assay, the authors assume the virulence proteins are secreted based on the size of the proteins. The molecular weight of key bands in the ladder are not provided, so the sizes cannot be appropriately confirmed by the reviewer. More importantly, the proteins present in the gel may not be the indicated virulence proteins secreted by the T3SS (see below). Western blot confirmation of at least one or two representative proteins should be performed, for example, using primary antibodies to *lpaB* and *lpaC* that are available. Finally, the SDS-PAGE gel appears overloaded. The proteins should be re-analyzed and the running conditions changed to improve the gel with appropriate separation of the proteins.

a. It is important to note that while Congo red was used to induce virulence protein secretion by the T3SS, it was added to the growing culture as mentioned in the materials and methods. Thus, both T3SS and proteins secreted by other mechanisms will be present in the supernatants that were analyzed by SDS-PAGE. Congo red secretion assays typically culture the bacteria first, then resuspend bacterial pellets in PBS with or without Congo red to induce secretion of virulence proteins (for example, see *Mol Microbiol.* (2013) 88(2):268-82. doi: 10.1111/mmi.12185). This process will ensure only T3SS proteins are present in the supernatants of supernatants treated with Congo red and will enable better visualization of protein bands at the appropriate sizes.

4. For the human neutrophil infections, the authors should provide the human subjects protocol number to ensure approved protocols from the institution.

Reviewer #3

(Remarks to the Author)

This is a well written and rigorous study examining genetic and phenotypic adaptations that help explain the observed epidemiological evidence for increased incidence of *Shigella sonnei* among human clinical shigellosis cases. The authors present an interesting and important finding that lineage 3 *S. sonnei* have evolved to resist host defenses, thereby providing support for their overall thesis that there are genetic and phenotypic changes that can explain why lineage 3 isolates are now predominant among human clinical infections. The overall presentation is very good. The authors include a well characterized strain from lineage II that is genetically and phenotypically different than the now predominant lineage 3

strains; this is important because many studies draw conclusions from this strain, but its continued use given the changing epidemiology of shigellosis is questionable. The inclusion of multiple strains for several assays (Figure S1) adds to the robustness of their findings, supporting their overall conclusions that lineage 3 strains have evolved to resist neutrophil-mediated stresses. As with essentially every paper, there are some points that should be clarified and discussed to allow for a more complete understanding of the results in the context of the field.

Minor Suggested Edits/Analyses:

Line 65: This is very picky – but the genus *Mycobacterium* is not human restricted (same is true for *Bordetella*). The authors are correct to specify *S. enterica* serovar Typhi as a human-restricted serovar of *Salmonella*, but then the genus is listed for *Mycobacterium* and *Bordetella*. To be consistent, I would recommend specifying the species *tuberculosis* and *pertussis*.

Line 82: Are *S. flexneri* and *S. sonnei* still referred to as different species? In line 49 the authors remark that “*Shigella* represents a group of human-adapted lineages of *Escherichia coli*”; therefore, this calls into question the use of “species” here to delineate *S. flexneri* and *S. sonnei*. This is semantical, and I appreciate that *Shigella* was only recently reclassified as a complex of *E. coli*, but perhaps now there is an opportunity for the authors to propose a better terminology than ‘species’ to refer to ‘flexneri’ vs. ‘sonnei’ or at least justify why the continued usage of ‘species’ here is justified for public health/infectious disease audiences.

Lines 115-116: Please see comments below. ABRicate is a great program and the methods used are good, my issue is that the interpretation of ‘absent’ may not be accurate here unless the authors can confirm that the genes are actually not present.

Line 133: I am assuming that 32.5C was used because the zebrafish cannot be cultured above this temperature. Still, with human body temperature being 37C (even higher during an infection), can the authors comment on how virulence gene expression of *S. sonnei* might be different at 37/38C to justify the use of this temperature. Again, I appreciate that this is likely due to the fish larvae not being able to withstand that high of a temperature, but as the authors are proposing that this is a good model to use, the difference in temperature and its effect on virulence gene expression is important to address here as a potential limitation of this model system.

Line 148: As strain 2.8 was elected to represent the other lineages for the dissemination experiment, that should be reported in the text. This may be a fair extrapolation, but since it was not tested, it is recommended to report ‘compared to the lineage 2.8 strain, representing non lineage 3 isolates, ...’

Line 201-202/Figure 2SA-B: In the figure, the authors compared survival due to infection with each lineage (2.8 vs. 3.6 vs. 3.7, all DMSO treated). I wonder if a better comparison would instead be to compare survival after challenge with the same strain, using DMSO vs. DXM treatment as the comparator? For example, if the authors compare infection with 3.6.1 with DMSO vs. DXM treatment, do they see a difference? This would more clearly show that it is reduction of inflammatory response (due to dexamethasone treatment) that is improving survival.

Figure 4E: recommend changing colors representing each gene. It looks the same as the colors used for lineage 3.7.29.1.4 used throughout the manuscript and at first glance it could be interpreted that these genes are unique to the lineage 3.7 strain. I think that the authors were trying to maintain the color scheme, but it is slightly confusing in the context of this figure and also Figure S4B. Could they instead use a shade of blue that is more different than the shade used for lineages 3.6 and 3.7 strains?

Line 235: I think this statement should be clarified a bit “likely explain differences in stress tolerances”; ‘stress tolerance’ encompasses many different things. Can the authors specify a bit about which stresses the lineage 3 are more tolerant to?

Line 271: In regard to a ‘shift towards a more host-restricted lifestyle’ – are the authors trying to communicate that this shift happened relatively recently (in reference to the epidemiological dominance of lineage 3, lines 272-273)? This is interesting given that *Shigella* is human-restricted (as the authors mention on line 281-282). It would be helpful to clarify this point a bit in the context provided in lines 270-281.

Line 296: It is recommended to add ‘model’ after zebrafish. Again, the thesis statement is that *Shigella* are human-restricted; zebrafish are used as a model here.

Figure 2 caption Line 319: should be ‘or’? ... infected with larvae at either 6 or 24 hours?

Lines 430-432: ABRicate utilizes blast to search for genes. Blast uses a threshold (and necessarily so!); genes that are either too short or too divergent (below XX% identity with reference) will not be included in the final output. For the authors to say that these genes are truly absent, I would recommend visually inspecting the genome at this region since they have whole genome sequences available and looking for hypothetical genes/coding sequences. The easiest way to do this is to look at the annotated genome files (.gff or similar) and use a genome browser to see whether the annotation software predicts that there is a gene present. This could also further support the authors’ hypothesis that degradation/loss of certain genes has an impact on immunogenicity. Otherwise, the authors need to rephrase their statements about gene ‘absence’ to ‘not detected.’

Lines 512-514: Why were the bacteria not removed (centrifugation and removal of media) prior to lysis?

Line 584: Please confirm that this was done at 59C and not 56C.

Version 1:

Reviewer comments:

Reviewer #1

(Remarks to the Author)

The authors addressed most of my concerns and clearly explained the technical limitations at this time that prevented addressing the few remaining points.

However, I believe that considering the rebuttal Figure 2 (not included in the revised manuscript), authors should reconsider

Fig. 5D (growth curves at pH5) and associated text.

As requested (major point 3), authors performed growth curves at pH5 and pH7 on all strains (16 in total). The result presented in rebuttal Figure 2 is difficult to fully appreciate due to the high number of strains but i) it does not support an increased tolerance to pH5 stress for lineage 3; ii) it is even not certain that it reproduces the result shown in Fig 5D.

Therefore, to my opinion, the statement L232-234 is not supported by the data ("In agreement, we found that Lineage 3 isolates grew more efficiently in acidic conditions (although Lineage 3.6 was intermediate between 53G and Lineage 3.7) (Fig 5C-D), a property which may confer enhanced resistance to phagosomal killing."). Furthermore, considering the results of rebuttal Figure 2, the authors should not keep Fig. 5C-D which only presents 3 strains (1 strain from lineage 2 and 2 strains from lineage 3) but should present a more complete figure. To make the figure clearer, I would suggest reducing the number of strains to 8: 2 strains from lineage 1, 2 strains from lineage 2 (including 2.8) and 4 strains from lineage 3 (including 3.6.1.1.1, 3.7.29.1.4 and 2 other strains with a clear increased virulence phenotype like 3.6.1 and 3.7.16).

The authors should be cautious in their conclusion. If the growth difference between lineage 3 and the other lineages at pH5 is not robust, the result could also be removed from the study. In this regard, in response to my main point 1, the authors tested bafilomycin on infected embryos, as shown in rebuttal Figure 1 (not included in the revised manuscript). If an increased acid stress resistance of lineage 3 were related to increased virulence, one would expect a lower mortality rate for lineage 3 strains in the presence of bafilomycin. However, bafilomycin had no effect on the mortality rate of the two lineage 3 strains tested (rebuttal Figure 1A). This result does not support a link between acid stress tolerance and enhanced virulence.

Reviewer #2

(Remarks to the Author)

The authors have appropriately addressed the critiques from the original manuscript and have included updated data and text modifications to address the concerns of the reviewers. The revised manuscript is improved and well-written with robust and reproducible data. Complementary methodologies are used to validate findings, and the appropriate number of replicate experiments and statistical analyses are used for each experiment. The study enhances our understanding of *Shigella sonnei* infection and demonstrates the usefulness of incorporating the zebrafish model into *Shigella* studies.

Reviewer #3

(Remarks to the Author)

The additional clarifications and experimental and data analyses are helpful and further support the authors' conclusions. Thank you for taking the time to address my comments.

REVIEWER COMMENTS

Reviewer #1 (Remarks to the Author):

The manuscript from Miles et al. addresses virulence phenotypes of *Shigella sonnei* lineages that are well distributed in humans, lineage 3 being the most frequent among the 5 *S. sonnei* lineages. The study initially focuses on lineages 1-3, before a detailed analysis on two strains of lineage 3 that are compared to the well-known 53G strain (of lineage 2). *S. sonnei* strains of lineage 3 harbour fewer genes encoding adhesion and fimbriae. The virulence of strains was assessed using zebrafish embryo as infection model at an unusual temperature (32,5°C), which allowed to discriminate lineage 3 from lineage 2. Lineage 3 strains are more virulent in zebrafish, where they exhibit a greater neutrophil response. In human neutrophils, lineage 3 strains exhibit greater tolerance to phagosomal killing. In addition, increased stress tolerance (to acid stress and complement) was observed for lineage 3 in vitro, as well as increased expression of group 4 capsule (G4C) genes. Authors conclude that cumulatively, these features may explain the epidemiological success of lineage 3 of *S. sonnei*.

A better characterization of *S. sonnei* pathogenicity/epidemiology is of major interest. Whereas the study provides an appealing new picture on *S. sonnei* lineage 3, few additional experiments are required to strengthen the findings and the conclusions. Several interesting observations are rather mild and lack experimental demonstration for causal links, which limits the impact of the conclusions (as proposed L265-273, L276-281, L285-292).

We thank the reviewer for their positive feedback and interest in our work. We performed additional experiments and included further clarifications to support our findings and strengthen our conclusions.

Major points

1) *S. sonnei* strains of lineage 3 harbour fewer genes encoding immunogenic components (adhesin and fimbriae) and express slightly more G4C. However, no experimental evidence supports that these feature have a causal link with higher virulence in zebrafish / enhanced stress tolerance / increased resistance to human neutrophils. Moreover, the link between stress tolerance and virulence phenotypes is not addressed.

Additional experiments should be considered to address these points. For example, since a *S. sonnei* 53G Δ tag4c is used in the study, a similar mutant could be constructed in a lineage 3 strain ? It would allow to address the contribution of G4C in some phenotypes.

We agree that linking virulence and stress tolerance is exciting. We believe an ideal experiment would be to make lineage 1 more like lineage 3 by ablating immunogenic components, or to re-introduce immunogenic components into lineage 3. However, in both cases such gain of function experiments for *Shigella* would not be ethical to advance in the UK in the absence of specialised licenses.

Future work is required to systematically investigate all lineage-specific differences, including loss of immunogenic components. Furthermore, we believe that variations in G4C alone are only likely to account partly for the lineage-dependent phenotypes observed here. Despite this, we have been trying to generate a G4C mutant in a lineage 3 background since August 2024. Major challenges in manipulating clinical isolates of *S. sonnei* (summarised in Scott et al, Nature Reviews Microbiology, 2025) include (1) the extreme and consistent loss of the

virulence plasmid (eg McVicker and Tang, Nature Microbiology, 2017; Martyn et al, Journal of Bacteriology, 2022), and (2) high levels of antibiotic resistance which prevent the use of available selectable markers (eg Hawkey et al, Nature Communications, 2021; summarised in Baker & Scott, Nature Reviews Microbiology, 2023).

During the revision process, we reached out to Dr. Abigail Clements (Imperial College London) who shared with us additional constructs (pSEVA Δ G4C and pSEVA Δ G4C-Kan, both of which are designed to replace the whole capsule locus) along with a triparental conjugation protocol to generate mutants in *S. sonnei*. Despite ongoing efforts, Dr. Clements highlighted that her work with Dr. Xosé Matanza has similarly not yet been able to generate a capsule mutant in lineage 3 clinical isolates whilst retaining the virulence plasmid (personal communication).

To seek out alternative methods of mutagenesis in response to the reviewer, we recently started a collaboration with Dr. Andrea Puhar and Dr. David Cisneros (Queen's University Belfast) to generate a G4C mutant in our clinical isolates using a novel cytosine base editing approach, which introduces premature stop codons into genes of interest (Sharma et al, mSystems, 2023).

While this work is ongoing, both teams involved in mutant generation are included as authors in the revised manuscript.

To assess the impact of higher tolerance to acid stress on lineage 3 strains in neutrophils / virulence phenotypes, authors should treat human neutrophils and zebrafish with an inhibitor of acidification (as bafilomycin). If the inhibition of acidification suppresses the difference between lineage 3 and lineage 2 strains (CFUs for neutrophils and embryo survival for zebrafish), it would support a link between acid stress tolerance of lineage 3 and increased tolerance to phagosomal killing / increased virulence.

Due to logistical issues in obtaining human neutrophils, we tested the impact of bafilomycin treatment on infected zebrafish (**Rebuttal Figure 1**). In agreement with the reviewer's hypothesis, we found that treatment with bafilomycin mildly suppressed differences between survival of larvae infected with lineage 2.8 and 3.6 *S. sonnei*. In the case of lineage 3.7 we saw no difference between treated and untreated embryos. These results may suggest that increased tolerance of lineage 3 is unaffected by bafilomycin treatment (e.g. because of increased capsule expression) and/or there are additional lineage-specific factors that drive stress tolerance.

Considering that treatment of zebrafish with bafilomycin has pleiotropic effects we prefer not to conclude on these experiments at this time.

Rebuttal Figure 1. The survival of zebrafish embryos infected with lineages 2.8, 3.6.1.1.1 and 3.7.29.1.4 treated with either 100µg/mL bafilomycin (BAF) or an equal concentration of DMSO (A). CFU counts from infected larvae at 0, 6 and 24 hours post infection (hpi).

2) CFUs from zebrafish. The profiles of time 0 are identical in panels C and D of Fig 1: if they represent inoculum, and not initial CFUs from infected fish, they should be removed from the graph. Whereas CFUs do not differ between strains at 24 hpi at 28,5°C, the authors make the same conclusion at 32,5°C, whereas the data suggest that there may be different trends (Fig 2D). However, the number of embryos tested at 24 hpi in the condition 32,5°C is too low (5 embryos), probably due to the high mortality rate. More embryos should be infected to have at least 12-15 embryos alive at 24 hpi for CFUs quantification. This is required to conclude if there is a difference or not in CFUs between the 4 strains at 24 hpi. In addition, authors speculate that lineage 3 of *S. sonnei* better withstand the bactericidal effect of the host immune response: how do they confront this hypothesis with the conclusion that no difference in CFUs are observed (Fig. 2D) ? An increased CFU number in zebrafish should be expected at some point with lineage 3 compared to lineage 2.

The data points in Figure 1C-D represent initial CFUs from the same infected zebrafish (since at 0 hpi, zebrafish have not yet been split into their respective temperature incubation groups). They are included on both graphs to allow for visual comparison to the initial CFU. This is clarified in the figure legend of the updated manuscript (line 355, page 14).

We repeated the 32.5°C condition to include at least 12 embryos at each time point, confirming that there is no significant difference in CFU burden between strains at either 6 or 24 hpi. Consistent with previous data from our first submission, a bimodal trend becomes clearer, likely representing embryos that will go on to clear the infection (with fewer CFUs) and those that will succumb at a later timepoint (with more CFUs). This updated figure is included in the revised manuscript (new Figure 2D). We note that these experiments are innately biased at later time points because we are strictly selecting for zebrafish that survived and will likely go on to clear infection (i.e. we cannot count bacterial burden from dead zebrafish because these data are not reliable). Therefore, this experiment may not be informative when investigating virulence phenotypes that present earlier.

We conclude that the difference in CFUs observed in human neutrophils is not quantifiable at the whole animal level due to complex interactions between different cell types *in vivo*. We are

specifically stating that neutrophil interactions and subsequent inflammation drive observed differences (supported by a new statistical analysis of dexamethasone experiments, see response to reviewer 3, page 11). In our whole animal system, the presence of other bactericidal impacts (such as macrophages and other components of the innate immune system) may mask differences in total CFUs observed from investigation of neutrophils in isolation. We clarify this in the updated Discussion (line 308, page 12).

3) Stress tolerance: the differences between lineage 2 and 3 are rather mild (not significant for 3.6 isolate in Fig 5A) and no statistical analysis in Fig 5D. These experiments are simple and should be strengthened by extending the analysis (growth at pH 5 should be evaluated on a longer time period, at least 6-7 hrs) and conducting experiments on all strains tested for virulence in Fig 2 and Fig S1 to have a broader / clearer picture of the impact of the results and justify the title.

As requested by the reviewer, we performed new experiments testing growth curves for all 16 strains for longer time period (ending at 5.5hr when bacteria reach stationary phase) but found a difference in baseline growth rates (**Rebuttal Figure 2**), likely due to the different metabolic capacities between lineages that we recently identified, <https://doi.org/10.1101/2025.02.03.635664>. We conclude that our experimental conditions tested here lack sensitivity for dissecting subtle stress responses across this many isolates when baseline growth differs across strains.

Rebuttal Figure 2. Bacterial strains were grown in TSB at either pH5 or pH7 and OD600 measurements were taken every 45 minutes.

4) Expression of G4C genes (Fig. 5F and 5H): the effect on *etk* and *etp* expression is mild and not significantly different for *etp*. Other genes of the operon (as *gfcA*, which is also implicated in the formation of the capsule) could be tested as well. In addition, the study could be extended to few other strains (at least 1 from each lineage ?) to strengthen the results and drive a conclusion. The conclusion L233-235: “Together, this establishes lineage-dependent variations in the expression of G4C, which likely explain observed differences in stress tolerance” is not supported by the data since a causal relation is not demonstrated (this would imply mutating G4C encoding genes in a lineage 3 strain for example).

We thank the reviewer for this suggestion. We repeated this experiment to include a lineage 1 strain and extended the qPCR panel to include an additional 3 genes (*gfcA*, *gfcB* and *gfcC*). The expression of the G4C genes was consistently higher in lineage 3.7, supporting our original hypothesis. Notably, these experiments again revealed that lineage 3.6 exhibits an intermediate expression profile between lineage 2.8 and 3.7, consistent with its behaviour in phagosomal killing assays (Figure 6F). The manuscript has been updated with these new data (page 10, new Figure 5F).

See response to point 1 with respect to mutant generation.

5) Phenotypes in human neutrophils (Fig 6): the two isolates of lineage 3 showed an increased propensity to survive within neutrophils. If this is also the case in the zebrafish model, again one would expect higher CFUs for lineage 3 strains at some point. In addition, authors could take advantage of the Tg(mpx::eGFP) fish line (Fig. 4D) to visualize the interaction between neutrophils and red fluorescent *S. sonnei* strains (as used in their previous publication, reference 33) at 6 hpi and 24 hpi.

See response to point 2 with respect to CFUs.

In reference 33 the experiment used mpeg1::Gal4-FF)gl25/Tg(UAS::LIFEACT-GFP)mu271 (a transgenic zebrafish line expressing GFP macrophages) and was performed to demonstrate that *S. sonnei* resides in macrophages throughout persistent carriage (Torraca et al, Journal of Infectious Diseases, 2023). In our case we are studying neutrophils. Moreover, we are not testing if *S. sonnei* can occupy the neutrophil niche long-term and instead we are showing that it can resist phagosomal killing.

While putting together this manuscript, we generated RFP transgenic strains of *S. sonnei* but observed that expression of RFP can reduce virulence in the case of clinical isolates, even though lab strain 53G is not significantly affected (our unpublished observations). As a result, it would not be reliable at this time to compare results from RFP strains with results from WT bacteria that all other experiments were performed with.

Minor points

1) The review “Pathogenicity and virulence of Shigella sonnei: A highly drug-resistant pathogen of increasing prevalence. Matanza XM, Clements A. Virulence. 2023. doi: 10.1080/21505594.2023.2280838.” should be cited

This review is now included together with other recent *S. sonnei* reviews (Scott et al, Nature Reviews Microbiology, 2024 and Miles et al, Trends in Microbiology, 2024) (line 89, page 4).

2) Survival curves: the higher virulence of lineage 3 is based on the analysis of 2 representative strains (Fig 2). Additional lineage 3 strains are tested in Fig S1 but Fig S1D is difficult to read due to a high number of strains (at time 48 hpi only 6 points out of 7 are visible). It seems that some strains of lineage 3 have profiles close to lineage 2 (strain 3.4 in panel C and 3.7.30.4.1 in panel D ?). It could be of interest to see how these two strains behave for other phenotypes ?

We agree that variability of some lineage 3 genotypes is an interesting phenotype, and this will inspire a significant line of future research in the Mostowy and Holt labs.

We improved readability in Figure S1.

3) Fig 4D: The authors should insist on the fact that the higher recruitment of neutrophils with lineage 3 is transitory (visible at 6hrs but not at 9h).

Text has been updated (line 207, page 8).

4) Zebrafish infection experiments are known to show some variability, as seen with the survival curves of Fig 2B and S2A. In figure S2A, the difference between lineage 3 and 2 is clear only after 24 hpi (which is not the case in figure 2B). Similarly, the differential effect of dexamethasone is only visible between 24 and 48 hpi (before 24 hpi, a similar reduced virulence is observed for lineages 2 and 3). How do the authors explain these findings considering the timing of the neutrophil response (6 hpi) ?

Minor variability in zebrafish infection experiments is expected to arise from infecting different clutches of embryos that may have slightly different immune responses (zebrafish like humans are outbred, unlike mice which are inbred).

In regard to the timing of the neutrophil response, we conclude that increased inflammation present at 6 hpi will control survival at later time points. See also response to point 2 with respect to lack of CFU differences,

We state these findings in the revised manuscript (line 211, page 9).

5) Strain 53G (2.8) was previously reported by the same team to establish persistent infection in zebrafish at 28°C. Do the virulent strains from cluster 3 also establish a persistent infection ? If not tested, this could be at least discussed ?

Two strains from lineage 3 were included in the manuscript describing persistent *S. sonnei* infection (Torraca et al, Journal of Infectious Diseases, 2023). Torraca et al found that *S. sonnei* lineages tested can persist equally well, although this work was not exhaustive. Importantly, this work was focussed on a macrophage niche, and not a neutrophil response as we are studying here.

Dr. Torraca has left the Mostowy lab and is now following up on *Shigella* persistent infection in his own lab at King's College London in the Department of Infectious Diseases.

L110-111: please clarify

Text has been updated (line 116, page 5).

Fig 1: the name of the genes that show differences among lineage 3 strains should be enlarged (difficult to read). The 4 strains mainly tested in the study should be easily identified (bold ?)

Figure 1 has been updated with increased font sizes, bolding of strains tested in the study and bolding of differences that are specific to Lineage 3 (new Figure 1).

L147: significant only for one of the two strains

Text has been updated to reflect this (line 165, page 7).

L156-157: the replication rate could make a difference for pINV stability in vitro / in vivo, thus this does not necessarily support that pINV is a key driver of *S. sonnei* in vivo

We acknowledge this point and updated text accordingly (line 176, page 7).

L201 (or Mat & Met): a reference with the use of DMX in zebrafish should be added

The methods section has been updated to include the following reference: Tobin, D. et al, Host Genotype-Specific Therapies Can Optimize the Inflammatory Response to Mycobacterial Infections, Cell, Volume 148, Issue 3, 434 – 446 (line 497, page 21).

L229: write “group 4 capsule” ?

The use of the G4C abbreviation is introduced on line 92, page 4.

L283: a specific reference for CHIM should be provided (rather than 2 reviews)

Text has been updated to include the following reference: Frenck Jr., R.W, Establishment of a Controlled Human Infection Model with a Lyophilized Strain of *Shigella sonnei* 53G, mSphere 5:10.1128/msphere.00416-20 (line 316, page 12).

L281-284: In addition to the CHIM, authors could also mention that it would be of interest to translate the virulence profiles obtained in zebrafish in the NAIP-NLRC4 deficient mouse model

Text has been updated (line 315, page 13).

Reviewer #2 (Remarks to the Author):

Summary: The goal of the manuscript was to compare different lineages of *Shigella sonnei* to identify factors that contribute to the infection dominance and epidemiological success of certain *S. sonnei* isolates. The study uses isolates from three separate lineages of *S. sonnei* and includes lab strains *S. sonnei* 53G and *S. flexneri* strain M90T for comparisons. The authors utilized comparative genomics to identify virulence genes harbored by the isolates and examined infection and dissemination in a zebrafish model. Subsequently, analyses were performed to further examine how certain *S. sonnei* isolates could effectively infect the model; and thus, examined stability of the virulence plasmid with secretion of virulence proteins, measured macrophage and neutrophil responses in the zebrafish, and assessed tolerance to stresses and phagosomal killing. Based on the data, the authors conclude that Lineage 3 isolates are more virulent and are more tolerant of stresses, which is related to upregulation of group four capsule synthesis genes. Combined with the epidemiological data of Lineage 3 isolates, the increased virulence and stress tolerance data provide reasons for the global spread of this lineage. Overall, the study enhances our understanding of *S. sonnei* infection and demonstrates the usefulness of incorporating the zebrafish model into *Shigella* studies.

Overall, the authors provide a well-written manuscript with reproducible data, utilize complementary methodology to validate their findings, and demonstrate appropriate number of replicate experiments and statistical analyses for each experiment. The data are convincing, but review of the manuscript identified some concerns that should be addressed by the authors. These concerns are outlined below.

We thank the reviewer for their supportive and thoughtful comments. We addressed all comments with new experiments and updated text.

1. For the genome comparisons, some clarifications would be helpful or are needed:

a. While M90T is a common and appropriate *S. flexneri* strain for comparison purposes, other *S. flexneri* isolates like serotype 2a strains harbor additional virulence genes and pathogenicity islands that are highly immunogenic or modulate the host immune response. These genes include, but are not limited to, *pic*, *sepA* and *sigA*. It would be useful to know if any of the *S. sonnei* isolates have acquired these genes. Further, consideration of these other virulence genes when comparing the isolates may help to determine potential roles for these virulence genes in *S. sonnei* given the data the authors obtained.

All canonical *E. coli* and *Shigella* virulence factors (including *pic*, *sepA* and *sigA*) are included in the ABRicate virulence gene database (ecoli_VF) that we originally used for screening (<https://github.com/tseemann/abricate>) and only those included in Figure 1 were detected in the isolates we screened. However, we appreciate the reviewer's comment and included a clarifying statement to rule out the presence of any other canonical *Shigella* virulence genes (line 115, page 5).

b. Regarding the adherence genes, the authors list genes that are absent in *S. sonnei*. For example, *fimBHGF* are absent in lineage 3 (as noted on line 119). These genes are relatively minor with regards to type 1 fimbriae expression as *fimB* encodes a recombinase for phase variable switching and *fimHGF* encode minor subunits of the fimbrial structure. It is unclear if critical genes such as *fimA* encoding the major subunit, *fimC* encoding the chaperone, and *fimD* encoding the pore/usher are present.

Likewise, for curli and based on the wording of the manuscript, it is unclear if the major subunit *csgA* or the pore gene *csgG* are present. If these genes have been maintained, it is possible the proteins are expressed. As demonstrated in the cited mSphere (2019) 4(6):e00751-19, doi: 10.1128/mSphere.00751-19, *S. flexneri* expresses adherence factors despite missing or non-functional genes.

In agreement, we clarified completeness of the *fim* and curli operons. We visually inspected the *fim* operons of lineage 3 genomes and found that a ~9.3 kb region (located between *gntP* and *nanC*, encoding genes *fimB*, *fimD*, *fimF*, *fimG* and *fimH*) is absent from all lineage 3 genomes, consistent with previous work showing degradation of the *fim* operon in *S. sonnei* reference strain Ss_046 (Bravo et al, PLoS One, 2015). We note that genes annotated as *fimA*, *fimB* and *fimC* are present in multiple copies at various points in each genome but are not associated with the 9.3 kb region lost in lineage 3 genomes and therefore likely represent homologs associated with different fimbrial clusters. To this end, we believe it is unlikely that type one fimbriae is expressed, due to the disjointed locus and absence of the fimbrial usher *fimD*.

Examination of the curli operon revealed that whilst *csgB* and *csgD* were not detected by ABRicate, other curli subunits, including *csgA* and *csgG* were indeed present in lineage 3 genomes. However, we observed the presence of an IS600 element within *csgD* in all lineage 2 and 3 genomes, making its expression unlikely (Ogasawara et al, Journal of Bacteriology, 2011). We now explicitly state that despite having lost some genes, it is possible that curli is expressed. These new analyses have been included in the revised manuscript and the relevant citation is included (line 126, page 5/6).

2. For the zebrafish model, the authors do not provide uninfected controls, especially for analyses performed at 32.5 C. Since zebrafish are typically cultured between 26-28.5 C, it is possible that the zebrafish were affected or even in some instances killed at 32.5 C. Zebrafish killing due to temperature will affect interpretation of the results. Finally, it would be helpful if the authors provided more context for temperature-regulated virulence in *Shigella*. A helpful citation is Front Mol Biosci. (2016) 3:61. doi: 10.3389/fmolb.2016.00061 that explains that virulence activation does start above 32 C.

Zebrafish are well documented to inhabit a wide range of temperatures in the wild, from as low as 6°C up to 38°C (Spence et al, Biological Reviews, 2008). The 32.5 C temperature is widely recognised as valuable for zebrafish infection studies involving thermoregulation (reviewed in Miles et al, Trends in Microbiology, 2024) and has been previously used in zebrafish infection studies, by our team (e.g. Torraca et al, PLOS Pathogens, 2019; Miles et al, mBio, 2023) and by other zebrafish teams (e.g. Brudal et al, Infection and Immunity, 2014; Zhang et al, Scientific Reports, 2022).

Appropriate controls have been included in our previous work (Torraca et al., PLOS Pathogens, 2019), showing that PBS injected larvae incubated at 32.5°C have a 0% rate of mortality. This information is included in the revised manuscript (line 148, page 6).

We also updated the text to highlight the thermoregulation of *Shigella* virulence and include the reviewer's helpful citation (line 149, page 6).

3. For the Congo red secretion assay, the authors assume the virulence proteins are secreted based on the size of the proteins. The molecular weight of key bands in the ladder are not provided, so the sizes cannot be appropriately confirmed by the reviewer. More importantly, the proteins present in the gel may not be the indicated virulence proteins secreted by the T3SS (see below). Western blot confirmation of at least one or two representative proteins should be performed, for example, using primary antibodies to IpaB and IpaC that are available. Finally, the SDS-PAGE gel appears overloaded. The proteins should be re-analyzed and the running conditions changed to improve the gel with appropriate separation of the proteins.

We improved the quality of the gel and provided the molecular weight markers for key bands (new Figure 3F). The molecular weights of known T3SS effector proteins are well-characterized, and the observed bands align with these expected sizes (Parsot et al, Molecular Microbiology, 1995). Furthermore, inclusion of a Congo Red-negative control for each lineage strengthens our interpretation that observed proteins are indeed T3SS effectors.

a. It is important to note that while Congo red was used to induce virulence protein secretion by the T3SS, it was added to the growing culture as mentioned in the materials and methods. Thus, both T3SS and proteins secreted by other mechanisms will be present in the supernatants that were analyzed by SDS-PAGE. Congo red secretion assays typically culture the bacteria first, then resuspend bacterial pellets in PBS with or without Congo red to induce secretion of virulence proteins (for example, see Mol Microbiol. (2013) 88(2):268-82. doi: 10.1111/mmi.12185). This process will ensure only T3SS proteins are present in the supernatants of supernatants treated with Congo red and will enable better visualization of protein bands at the appropriate sizes.

The protocol we followed (introduced in Reinhardt et al, BioProtocol, 2014) is described in the literature (Dohlich et al, PLoS Path, 2014; Pinaud et al, PLoS One, 2017). This experiment was incorporated to test whether the T3SS varies between strains and our experiments (both protein secretion and qPCR) show minimal variation (Figure 3E-F). The inclusion of Congo red negative controls rules out inclusion of proteins secreted by other mechanisms.

In future work to more precisely compare T3SS protein secretion, we will follow the reviewer's advice.

4. For the human neutrophil infections, the authors should provide the human subjects protocol number to ensure approved protocols from the institution.

Text has been updated (line 537, page 22)

Reviewer #3 (Remarks to the Author):

This is a well written and rigorous study examining genetic and phenotypic adaptations that help explain the observed epidemiological evidence for increased incidence of *Shigella sonnei* among human clinical shigellosis cases. The authors present an interesting and important finding that lineage 3 *S. sonnei* have evolved to resist host defenses, thereby providing support for their overall thesis that there are genetic and

phenotypic changes that can explain why lineage 3 isolates are now predominant among human clinical infections. The overall presentation is very good. The authors include a well characterized strain from lineage II that is genetically and phenotypically different than the now predominant lineage 3 strains; this is important because many studies draw conclusions from this strain, but its continued use given the changing epidemiology of shigellosis is questionable. The inclusion of multiple strains for several assays (Figure S1) adds to the robustness of their findings, supporting their overall conclusions that lineage 3 strains have evolved to resist neutrophil-mediated stresses. As with essentially every paper, there are some points that should be clarified and discussed to allow for a more complete understanding of the results in the context of the field.

We thank the reviewer for their enthusiasm.

Minor Suggested Edits/Analyses:

Line 65: This is very picky – but the genus *Mycobacterium* is not human restricted (same is true for *Bordetella*). The authors are correct to specify *S. enterica* serovar Typhi as a human-restricted serovar of *Salmonella*, but then the genus is listed for *Mycobacterium* and *Bordetella*. To be consistent, I would recommend specifying the species tuberculosis and pertussis.

We agree and clarified the revised text (line 71, page 3).

Line 82: Are *S. flexneri* and *S. sonnei* still referred to as different species? In line 49 the authors remark that “*Shigella* represents a group of human-adapted lineages of *Escherichia coli*”; therefore, this calls into question the use of “species” here to delineate *S. flexneri* and *S. sonnei*. This is semantical, and I appreciate that *Shigella* was only recently reclassified as a complex of *E. coli*, but perhaps now there is an opportunity for the authors to propose a better terminology than ‘species’ to refer to ‘flexneri’ vs. ‘sonnei’ or at least justify why the continued usage of ‘species’ here is justified for public health/infectious disease audiences.

We agree and replaced any use of ‘species’ (with reference to *Shigella*) to ‘subgroup’ (line 89, page 4).

Lines 115-116: Please see comments below. ABRicate is a great program and the methods used are good, my issue is that the interpretation of ‘absent’ may not be accurate here unless the authors can confirm that the genes are actually not present.

This point was also raised by reviewer 2 (point 1b). All uses of the word ‘absence’ in reference to gene loss have been replaced by ‘not detected’ (line 126, page 5/6).

Line 133: I am assuming that 32.5C was used because the zebrafish cannot be cultured above this temperature. Still, with human body temperature being 37C (even higher during an infection), can the authors comment on how virulence gene expression of *S. sonnei* might be different at 37/38C to justify the use of this temperature. Again, I appreciate that this is likely due to the fish larvae not being able to withstand that high of a temperature, but as the authors are proposing that this is a good model to use, the difference in temperature and its effect on virulence gene expression is important to address here as a potential limitation of this model system.

An incubation temperature of 32.5°C was selected as it represents a condition that is both safe for zebrafish maintenance (see response to reviewer 2, point 2) and above the temperature at which the T3SS is activated (32°C). However, we acknowledge that gene

expression profiles may differ at 37°C and address this in the discussion section (line 313, page 13).

Line 148: As strain 2.8 was elected to represent the other lineages for the dissemination experiment, that should be reported in the text. This may be a fair extrapolation, but since it was not tested, it is recommended to report ‘compared to the lineage 2.8 strain, representing non lineage 3 isolates,’

Text has been updated (line 165, page 7).

Line 201-202/Figure 2SA-B: In the figure, the authors compared survival due to infection with each lineage (2.8 vs. 3.6 vs. 3.7, all DMSO treated). I wonder if a better comparison would instead be to compare survival after challenge with the same strain, using DMSO vs. DXM treatment as the comparator? For example, if the authors compare infection with 3.6.1 with DMSO vs. DXM treatment, do they see a difference? This would more clearly show that it is reduction of inflammatory response (due to dexamethasone treatment) that is improving survival.

We thank the reviewer for this suggestion, and repeated statistical comparisons to compare groups. We found a statistical significance in survival between DMSO vs DXM treatment for lineages 3.6 ($p=0.021$) and for 3.7 ($p=0.042$), but not for lineage 2.8. These new analyses support our conclusion that increased inflammation contributes to the enhanced virulence of lineage 3 (new Figure S2A).

Figure 4E: recommend changing colors representing each gene. It looks the same as the colors used for lineage 3.7.29.1.4 used throughout the manuscript and at first glance it could be interpreted that these genes are unique to the lineage 3.7 strain. I think that the authors were trying to maintain the color scheme, but it is slightly confusing in the context of this figure and also Figure S4B. Could they instead use a shade of blue that is more different than the shade used for lineages 3.6 and 3.7 strains?

Colour scheme has been modified (new Figure 5E, new figure S5B).

Line 235: I think this statement should be clarified a bit “likely explain differences in stress tolerances”; ‘stress tolerance’ encompasses many different things. Can the authors specify a bit about which stresses the lineage 3 are more tolerant to?

Text has been updated to clarify that lineage 3 is more tolerant to complement mediated and phagosomal killing (line 257, page 10).

Line 271: In regard to a ‘shift towards a more host-restricted lifestyle’ – are the authors trying to communicate that this shift happened relatively recently (in reference to the epidemiological dominance of lineage 3, lines 272-273)? This is interesting given that *Shigella* is human-restricted (as the authors mention on line 281-282). It would be helpful to clarify this point a bit in the context provided in lines 270-281.

We propose that *S. sonnei* currently represents the least-adapted subgroup of *S. sonnei* and our findings suggest that it is on the same evolutionary trajectory towards host restriction, as has been seen in *S. dysenteriae* (which has undergone greater genome streamlining (Hawkey et al, PLoS Genetics, 2020) and is considered the most virulent subgroup of *Shigella* (Schnupf et al, Microbiology Spectrum, 2019)). The discussion has been updated to clarify this point (line 293, page 12).

Line 296: It is recommended to add ‘model’ after zebrafish. Again, the thesis statement is that *Shigella* are human-restricted; zebrafish are used as a model here.

Text has been updated (line 308, page 12).

Figure 2 caption Line 319: should be 'or'? ... infected with larvae at either 6 or 24 hours?

Text has been updated (line 354, page 14).

Lines 430-432: ABRicate utilizes blast to search for genes. Blast uses a threshold (and necessarily so!); genes that are either too short or too divergent (below XX% identity with reference) will not be included in the final output. For the authors to say that these genes are truly absent, I would recommend visually inspecting the genome at this region since they have whole genome sequences available and looking for hypothetical genes/coding sequences. The easiest way to do this is to look at the annotated genome files (.gff or similar) and use a genome browser to see whether the annotation software predicts that there is a gene present. This could also further support the authors' hypothesis that degradation/loss of certain genes has an impact on immunogenicity. Otherwise, the authors need to rephrase their statements about gene 'absence' to 'not detected.'

This point was also raised by reviewer 2 (point 1b). We investigated intactness of the operons of interest and discovered that some gene remnants are indeed present, as per the reviewer's hypothesis. Whilst we cannot predict gene functionality from this work, the overall trend of degradation (with regards to immunogenic components) is maintained following this new analysis. We now explicitly include these findings in the revised manuscript and remove any references to gene absence, instead using the term 'not detected' (line 126, page 5/6). We thank the reviewer for highlighting this important clarification.

Lines 512-514: Why were the bacteria not removed (centrifugation and removal of media) prior to lysis?

We followed an established protocol designed to prevent excessive neutrophil death due to an additional centrifugation post infection (Torraca et al, PLoS Pathogens, 2019). Our text has been updated accordingly (line 554, page 23).

Line 584: Please confirm that this was done at 59C and not 56C.

Heat inactivation was performed at 56°C, a temperature widely reported for inactivation of complement (for example, Soltis et al, Immunology, 1979; Mellors et al, Nature Communications, 2025).

Our text has been updated accordingly to include supporting references (line 601, page 25).

Reviewer #1 (Remarks to the Author):

The authors addressed most of my concerns and clearly explained the technical limitations at this time that prevented addressing the few remaining points.

We thank the reviewer for their constructive comments and feedback helping to strengthen our manuscript message.

However, I believe that considering the rebuttal Figure 2 (not included in the revised manuscript), authors should reconsider Fig. 5D (growth curves at pH5) and associated text.

As requested (major point 3), authors performed growth curves at pH5 and pH7 on all strains (16 in total). The result presented in rebuttal Figure 2 is difficult to fully appreciate due to the high number of strains but i) it does not support an increased tolerance to pH5 stress for lineage 3; ii) it is even not certain that it reproduces the result shown in Fig 5D.

Therefore, to my opinion, the statement L232-234 is not supported by the data (“In agreement, we found that Lineage 3 isolates grew more efficiently in acidic conditions (although Lineage 3.6 was intermediate between 53G and Lineage 3.7) (Fig 5C-D), a property which may confer enhanced resistance to phagosomal killing.”). Furthermore, considering the results of rebuttal Figure 2, the authors should not keep Fig. 5C-D which only presents 3 strains (1 strain from lineage 2 and 2 strains from lineage 3) but should present a more complete figure. To make the figure clearer, I would suggest reducing the number of strains to 8: 2 strains from lineage 1, 2 strains from lineage 2 (including 2.8) and 4 strains from lineage 3 (including 3.6.1.1.1, 3.7.29.1.4 and 2 other strains with a clear increased virulence phenotype like 3.6.1 and 3.7.16).

The authors should be cautious in their conclusion. If the growth difference between lineage 3 and the other lineages at pH5 is not robust, the result could also be removed from the study. In this regard, in response to my main point 1, the authors tested bafilomycin on infected embryos, as shown in rebuttal Figure 1 (not included in the revised manuscript). If an increased acid stress resistance of lineage 3 were related to increased virulence, one would expect a lower mortality rate for lineage 3 strains in the presence of bafilomycin. However, bafilomycin had no effect on the mortality rate of the two lineage 3 strains tested (rebuttal Figure 1A). This result does not support a link between acid stress tolerance and enhanced virulence.

Importantly, results from Figure 5D (previous submission) were reproduced in Rebuttal Figure 2 (previous rebuttal letter) with the original strains and timepoints tested (strains 2.8, 3.6.1.1, and 3.7.29.1.4).

However, from our experiments in Rebuttal Figure 2 (previous rebuttal letter) involving 16 different strains, we agree that differences were not robust across the entirety of lineage 3, especially in comparison to lineage 1. These results were confounded by differences in baseline growth rates and we speculate that this is partly due to predicted differences in metabolic capacity between lineages that we recently identified <https://doi.org/10.1101/2025.02.03.635664>.

We therefore agree with the reviewer to remove this result from the study and have modified text to remove any references linking acid stress and virulence. For this we updated line 48 (removed 'and acidic conditions' in Abstract), lines 232-234 (removed description of this result) and Figures 5C-D (removed Figures and corresponding Figure Legends) from our previous submission. Our revised submission has otherwise not changed and therefore does not contain any highlighted text.

Reviewer #2 (Remarks to the Author):

The authors have appropriately addressed the critiques from the original manuscript and have included updated data and text modifications to address the concerns of the reviewers. The revised manuscript is improved and well-written with robust and reproducible data. Complementary methodologies are used to validate findings, and the appropriate number of replicate experiments and statistical analyses are used for each experiment. The study enhances our understanding of Shigella sonnei infection and demonstrates the usefulness of incorporating the zebrafish model into Shigella studies.

We thank the reviewer for their enthusiasm.

Reviewer #3 (Remarks to the Author):

The additional clarifications and experimental and data analyses are helpful and further support the authors' conclusions. Thank you for taking the time to address my comments.

We thank the reviewer for their positive feedback.